# BézierFlow: Learning Bézier Stochastic Interpolant Schedulers for Few-Step Generation

**Yunhong Min**[*]   **Juil Koo**[*]   **Seungwoo Yoo**   **Minhyuk Sung**
KAIST
{dbsghd363,63days,dreamy1534,mhsung}@kaist.ac.kr

## Abstract

We introduce BézierFlow, a lightweight training approach for few-step generation with pretrained diffusion and flow models. BézierFlow achieves a 2–3× performance improvement for sampling with $\leq 10$ NFEs while requiring only 15 minutes of training. Recent lightweight training approaches have shown promise by learning optimal timesteps, but their scope remains restricted to ODE discretizations. To broaden this scope, we propose learning the optimal transformation of the sampling trajectory by parameterizing stochastic interpolant (SI) schedulers. The main challenge lies in designing a parameterization that satisfies critical desiderata, including boundary conditions, differentiability, and monotonicity of the SNR. To effectively meet these requirements, we represent scheduler functions as Bézier functions, where control points naturally enforce these properties. This reduces the problem to learning an ordered set of points in the time range, while the interpretation of the points changes from ODE timesteps to Bézier control points. Across a range of pretrained diffusion and flow models, BézierFlow consistently outperforms prior timestep-learning methods, demonstrating the effectiveness of expanding the search space from discrete timesteps to Bézier-based trajectory transformations. Project Page: https://bezierflow.github.io.

## 1 Introduction

Diffusion and flow models have achieved state-of-the-art performance, but at the cost of high computation due to their iterative generative processes. A large body of recent work (Lu et al., 2022; Song et al., 2023; Liu et al., 2023; Tong et al., 2025) has aimed to accelerate generation to just a few steps. For diffusion models, many dedicated solvers (Lu et al., 2022; 2023; Zhang & Chen, 2023; Zhao et al., 2023) tailored to their ODE formulations have been proposed. Although these methods substantially reduce the number of iterations from hundreds to tens, they are often insufficient to bring the steps down to only a few. More recent distillation techniques, such as consistency models (Song et al., 2023) and variants (Kim et al., 2024; Berthelot et al., 2023; Zhou et al., 2025) for diffusion and ReFlow (Liu et al., 2023) for flow models, can reduce the number of steps to as few as one, but they require substantial fine-tuning, often hundreds to thousands of GPU hours, even for small datasets.

A notable line of recent work is the lightweight training approach, which learns only a few parameters with a pretrained model to improve output quality for a small number of function evaluations (NFEs). Compared to the distillation techniques, such approaches require only tens of GPU minutes for training while achieving considerable improvements. The key questions for lightweight training are: (1) what to optimize, and (2) how to parameterize the variables. For the former, most previous work (Tong et al., 2025; Chen et al., 2024; Xue et al., 2024) has focused on learning the optimal sequence of timesteps for ODE solves, treating a nondecreasing sequence of timesteps as learnable variables. Most notably, a recent teacher-forcing approach (Tong et al., 2025) that uses the outputs of a multistep adaptive solver as the teacher has demonstrated the effectiveness of learned ODE timesteps.

---

[*]Equal contribution.

Broadening our scope, we explore variables beyond ODE timesteps that can be learned with lightweight training. As our first key contribution, we propose optimizing the sampling trajectories themselves. The Stochastic Interpolant (SI) framework (Albergo et al., 2023) provides a unified view of modern ODE-based generative models. In this framework, the state at any time is written as a linear interpolation between two endpoint samples: one drawn from the source (e.g., latent) distribution and the other from the target data distribution. The interpolation is governed by a pair of time-dependent coefficient functions, referred to as the SI scheduler. The scheduler fully specifies the geometry of the sampling trajectory. Different models adopt different schedulers, yet interchanging one scheduler for another at inference time does not change the endpoint marginal distributions. Inspired by this, we propose a lightweight training framework for learning an SI scheduler, which is equivalent to sampling path transformations that preserve the endpoints.

Our second key contribution lies in the parameterization of SI schedulers. The 1D continuous functions for SI schedulers must satisfy the following properties: (i) *boundary conditions*, ensuring that the endpoints of the coefficients are fixed, (ii) *monotonicity*, which guarantees a strictly nondecreasing signal-to-noise ratio (SNR) along the sampling path, and (iii) *differentiability*, which ensures that a velocity field can be derived in the ODEs governed by the learned scheduler. To effectively parameterize the space of such functions, while restricting the scope to polynomials, we propose a *Bézier*-based parameterization, termed the Bézier SI Scheduler, which forms the core of our overall lightweight training framework, BézierFlow. A 1D Bézier function naturally satisfies all of these properties: the boundary conditions can be enforced by simply setting the two end control points to the time range boundaries; the function is smooth and differentiable by the definition of polynomial Bézier curves; and monotonicity can be achieved by enforcing a nondecreasing order of control points, making the learning process identical to learning the ODE timesteps in previous work (Tong et al., 2025).

In our experiments, we evaluate both diffusion and flow models across diverse datasets and ODE solvers. BézierFlow consistently outperforms existing acceleration techniques while requiring only lightweight training, taking around 15 minutes on a single GPU. Extensive results further demonstrate the effectiveness of optimizing the sampling trajectories directly, rather than ODE timesteps, when coupled with our continuous Bézier-based parameterization.

## 2 RELATED WORK

There have been various attempts to improve few-step generation in ODE-based generative models. One major line of work focuses on designing dedicated ODE solvers tailored to the dynamics of the models (Lu et al., 2022; 2023; Zhao et al., 2023; Zhang & Chen, 2023). Although these methods require no additional training, they are unable to achieve high-fidelity generation with only a few steps. Another line is distillation-based approaches (Song et al., 2023; Salimans & Ho, 2022; Liu et al., 2023), which have demonstrated impressive gains in the very low-NFE regime, but incur substantial computational cost in training. Despite these diverse strategies, our lightweight training approach is most closely related to the methods listed below, which we now discuss in more detail.

**Learning ODE Solving Timesteps.** Several methods aim to achieve high-fidelity generation with few NFEs by optimizing the ODE timesteps in a lightweight manner. Chen et al. (2024) frame ODE timestep learning as a selection problem under a fixed NFE budget. Based on statistics collected from multiple sampling trajectories, they allocate more steps to regions of high curvature and fewer to flatter regions. Xue et al. (2024) optimize timesteps from the perspective of numerical integration: given a specific ODE solver, they minimize the accumulated local integration error along the trajectory. Tong et al. (2025) learn optimal timesteps through a data-driven distillation framework, where a high-NFE sampler serves as the teacher and a low-NFE sampler as the student. The timesteps are optimized by minimizing the discrepancy between their outputs starting from the same initial noise. Compared to these methods that learn optimal ODE timesteps, we learn optimal stochastic interpolant schedulers and demonstrate superior performance over these approaches.

**Learning Sampling Trajectories.** Several works (Karras et al., 2022; Lipman et al., 2024; Pokle et al., 2024; Kim et al., 2025) have explored changing sampling trajectories at inference time to improve generation quality and diversity, while selecting from a few predefined stochastic interpolant schedulers (e.g., linear, VP, VE), rather than parameterizing the function space and finding the best scheduler through optimization. To our knowledge, Bespoke Solver (Shaul et al., 2024) is the only

approach that learns an optimal sampling trajectory. Unlike our method, however, it relies on a discrete parameterization, which prevents direct derivation of first-order derivatives. These derivatives must instead be represented through auxiliary variables, introducing redundancy that can lead to inconsistencies between zeroth-order and first-order representations, and thus often fail to capture a truly differentiable function.

## 3 BACKGROUND: STOCHASTIC INTERPOLANT FRAMEWORK

Stochastic Interpolant (SI) (Albergo et al., 2023) is a unified framework for generative modeling, encompassing both ODE-based and SDE-based models (Song et al., 2021b; Ho et al., 2020; Song et al., 2021a; Lipman et al., 2023). Given two marginal probability densities $p_0, p_1 : \mathbb{R}^d \to \mathbb{R}_{\geq 0}$, a stochastic interpolant $x(t)$ is defined by the following stochastic process:

$$x(t) = \alpha(t)x_1 + \sigma(t)x_0 + \gamma(t)z, \quad t \in [0, 1], \tag{1}$$

where $\alpha(t)$ and $\sigma(t)$ are interpolation coefficients between $x_0 \sim p_0$ and $x_1 \sim p_1$, and $z \sim \mathcal{N}(0, I)$ is a latent variable introducing stochasticitiy. The process satisfies the boundary conditions $x(0) = x_0$ and $x(1) = x_1$ by enforcing $\alpha(0) = \sigma(1) = 0$, $\alpha(1) = \sigma(0) = 1$, and $\gamma(0) = \gamma(1) = 0$.

While the general formulation written in Eq. 1 is broad, many important generative models—including diffusion (Song et al., 2021a; Ho et al., 2020; Karras et al., 2022), flow (Lipman et al., 2023; 2024), and score-based (Song et al., 2021b) models—can be expressed in a more specific form, referred to as one-sided interpolants:

$$x(t) = \alpha(t)x_1 + \sigma(t)x_0, \tag{2}$$

where, as a common practice, $p_0 = \mathcal{N}(0, I)$ and $p_1 = p_{\text{data}}$, thereby the latent variable $z$ is absorbed into the initial state $x_0$. By differentiating both sides of Eq. 2, it can be expressed in the following ODE form:

$$\frac{dx(t)}{dt} = \dot{\alpha}(t)x_1 + \dot{\sigma}(t)x_0, \tag{3}$$

where we denote a time derivative by the dot. For these dynamics to be well defined, $\alpha(t)$ and $\sigma(t)$ must be twice continuously differentiable ($C^2$) to ensure that the divergence terms in the associated Fokker-Planck equation are well-defined.

Based on Eq. 2, within the SI framework, different ODE-based generative models, including diffusion, flow, score-based models, learn different but interchangeable quantities. Along the sampling path $(x, t)$, flow models $v_\phi(x, t)$ approximate the velocity field $u_t(x) = \mathbb{E}[\dot{x}_t \mid x_t = x]$, while diffusion models $\epsilon_\phi(x, t)$ approximate the expected initial random noise state $\eta_t(x) = \mathbb{E}[x_0 \mid x_t = x]$. Finally, score-based models $s_\phi(x, t)$ estimate the score function, which is equivalent to the scaled version of the expected initial state: $\nabla \log p_t(x) = -\sigma^{-1}(t)\eta_t(x)$. Thus, different types of generative models are mathematically linked, and under the SI framework, a pretrained model of one type can be reinterpreted as another at inference. For convenience, we collectively refer to these ODE-based generative models as the SI model, denoted by $S_\phi(x, t)$, throughout the paper.

## 4 BÉZIERFLOW

### 4.1 PROBLEM DEFINITION

The objective of our work is to learn an optimal *sampling trajectory* that enables high-quality generation with a few NFEs (e.g., $\leq 10$), while using a pretrained diffusion or flow model.

We consider two sampling trajectories that share the same endpoints $x_0$ and $x_1$. The source path refers to the trajectory used during model training, while the target path is a newly optimized trajectory for inference. Although both trajectories share the same endpoints, we assume their intermediate geometry matters: due to discretization error in ODE solving, output quality would depend on the path geometry. Given this assumption, we therefore aim to optimize the target path such that, even with only a few of NFEs, its geometry produces sampling results comparable to those obtained along the source path with many steps.

Formally, given a pretrained SI model $S_\phi$, let $\xi(x_0, \{t_i\}_{i=1}^N; S_\phi)$ denote a multistep ODE solver along the source path (the *teacher*), and $\bar{\xi}_\theta(x_0, \{s_i\}_{i=1}^M; S_\phi)$ a few-step ODE solver along the target

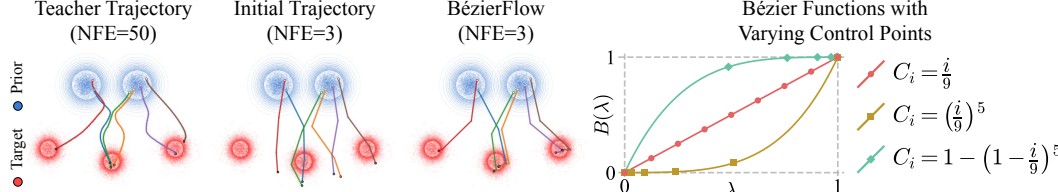

Figure 1: **Illustration of Sampling Trajectories and Bézier Functions.** On the left, we visualize different sampling trajectories. While initial trajectories deviate from the target distribution, BézierFlow aligns them with those of the teacher using only NFE=3. On the right, we present examples of 8-degree Bézier functions with different arrangements of control points.

path (the *student*), where $\{t_i\}_{i=1}^N$ and $\{s_i\}_{i=1}^M$ are the respective timestep sets with $M \ll N$. Although both solvers start from the same initial state $x_0 \sim p_0$, they differ in the number of NFEs and the sampling path. Let $q(x_1)$ denote the distribution induced by the teacher, and $\bar{p}_\theta(x_1)$ the student's output distribution. Our objective is formulated as the following teacher-forcing KL minimization:

$$\min_\theta D_{\mathrm{KL}}\left(q(x_1)\|\bar{p}_\theta(x_1)\right). \tag{4}$$

In practice, we optimize Eq. 4 using the following tractable surrogate objective (Tong et al., 2025), which enforces the outputs of the two solvers to align with each other:

$$\min_\theta \mathcal{L}(\theta) = \mathbb{E}_{x_0 \sim p_0}\left[d\big(\xi(x_0, \{t_i\}_{i=1}^N; S_\phi), \bar{\xi}_\theta(x_0, \{s_i\}_{i=1}^M; S_\phi)\big)\right], \tag{5}$$

where $d(\cdot, \cdot)$ is a distance metric such as LPIPS (Zhang et al., 2018). This lightweight optimization adjusts only the target scheduler coefficients while using the pretrained model, thereby improving few-step generation at minimal training cost. See the left of Fig. 1, which compares sampling trajectories from the prior distribution to the target distribution: (i) the teacher's trajectories with many steps (NFE=50), (ii) the student's initial trajectories, and (iii) trajectories optimized by BézierFlow. While the student's initial trajectories deviate from the target distribution at NFE=3, after training with BézierFlow, they closely follow those of the teacher despite the much smaller NFE.

## 4.2 SAMPLING PATH TRANSFORMATION

As discussed in Sec. 4.1, in order to define two trajectories that share the same endpoints, we view this as a transformation of the source path via the path reparameterization following prior works (Karras et al., 2022; Shaul et al., 2024; Pokle et al., 2024; Kim et al., 2025).

In the SI framework, a sampling path is governed by a pair of interpolation coefficients in Eq. 2, which we refer to as a *scheduler*. We denote the coefficients of the source path as the source scheduler $(\alpha_t, \sigma_t)$, and those of the target path as the target scheduler $(\bar{\alpha}_s, \bar{\sigma}_s)$. Specifically, to relate the two schedulers, we adopt a scaling reparameterization trick (Karras et al., 2022), where the target state $\bar{x}_s$ is defined from the source state $x_t$ as $\bar{x}_s = c_s x_{t_s}$. Here, $c_s$ can be arbitrary scalar functions and $t_s$ any invertible mapping from $s$, but we define them as

$$c_s = \begin{cases} \frac{\bar{\sigma}(s)}{\sigma(t_s)} = \frac{\bar{\alpha}(s)}{\alpha(t_s)}, & 0 < s < 1, \\ 1, & \text{otherwise}, \end{cases} \tag{6}$$

$$t_s = t(s) = \rho^{-1}\big(\bar{\rho}(s)\big), \qquad \rho(t) = \frac{\alpha(t)}{\sigma(t)}, \ \bar{\rho}(s) = \frac{\bar{\alpha}(s)}{\bar{\sigma}(s)}, \tag{7}$$

where both $\rho$ and $\bar{\rho}$ are invertible as the signal-to-noise ratio (SNR) increases monotonically over time.

Applying the change-of-variables to Eq. 3, the velocity in the target path $\bar{u}_s(\bar{x}_s)$ is expressed as

$$\bar{u}_s(\bar{x}_s) = \frac{d\bar{x}_s}{ds} = \left(\partial_s \log c_s\right)\bar{x}_s + c_s \frac{dt_s}{ds} u_{t_s}\left(\frac{\bar{x}_s}{c_s}\right). \tag{8}$$

We note that replacing the source scheduler with the target scheduler based on Eq. 6 at inference is valid for two reasons: (i) the endpoint marginal distributions are preserved, and (ii) the training

objective of a SI model is invariant to the choice of schedule as long as the SNR endpoints (minimum and maximum values) are the same. Thus, learning $(\bar{\alpha}_s, \bar{\sigma}_s)$ only changes the geometry of the sampling path, and hence the few-step discretization behavior, without altering the underlying target distributions or requiring a different pretrained SI model. See App. A for more details.

## 4.3 BÉZIER STOCHASTIC INTERPOLANT SCHEDULER

In Sec. 4.2, we discussed *what* to optimize, namely the sampling path determined by the SI scheduler $(\bar{\alpha}_s, \bar{\sigma}_s)$. The next crucial question is *how* to parameterize these 1D continuous functions effectively. Since the space of arbitrary 1D functions is prohibitively large, we employ 1D Bézier parameterization, which offers strong expressiveness with a compact number of parameters—the control points. Moreover, Bézier functions naturally satisfy the three key requirements of the SI scheduler: (i) boundary conditions, as described in Sec. 3, (ii) monotonicity to ensure a strictly nondecreasing signal-to-noise ratio (SNR), and (iii) differentiability to compute the transformed velocity in Eq. 8.

An $n$-degree Bézier curve is defined as a weighted linear combination of $n+1$ control points $\{C_i\}_{i=0}^n$, where the weights are given by Bernstein basis polynomials $b_{i,n}$:

$$B(\lambda) = \sum_{i=0}^n b_{i,n}(\lambda) C_i, \quad b_{i,n}(\lambda) = \binom{n}{i}(1-\lambda)^{n-i}\lambda^i, \quad \lambda \in [0,1]. \tag{9}$$

As shown in Eq. 9, with only $n$ control points, it can represent a wide range of trajectories. Unlike arbitrary 1D polynomial functions, Bézier functions always pass through their control points in order, making it straightforward to enforce boundary conditions and monotonicity. See the right of Fig. 1, which illustrates how 1D Bézier functions $B(\lambda)$, $\lambda \in [0,1]$ can represent diverse shapes under different control point arrangements while keeping the endpoints fixed.

Moreover, they are inherently smooth and infinitely differentiable $(C^\infty)$, with a closed-form derivative:

$$\dot{B}(\lambda) = n \sum_{i=0}^{n-1} b_{i,n-1}(\lambda)(C_{i+1} - C_i), \tag{10}$$

which allows us to directly compute the transformed velocity in Eq. 8 at any time $s$. Specifically, we parameterize $\bar{\alpha}(s)$ and $\bar{\sigma}(s)$ as $n$-degree 1D Bézier functions, each defined by a set of control points:

$$\bar{\alpha}^\theta(s) = (\alpha_1 - \alpha_0) \sum_{i=0}^n b_{i,n}(s) C_i^{(\alpha)} + \alpha_0, \qquad \bar{\sigma}^\theta(s) = (\sigma_1 - \sigma_0) \sum_{i=0}^n b_{i,n}(s) C_i^{(\sigma)} + \sigma_0. \tag{11}$$

For the boundary conditions, we fix the end control points $(C_0^{(\alpha)} = C_0^{(\sigma)} = 0, C_n^{(\alpha)} = C_n^{(\sigma)} = 1)$ and treat only the $n-1$ interior control points as parameters. Concretely, with learnable parameters $\theta^{(\alpha)}, \theta^{(\sigma)} \in \mathbb{R}^{n-1}$, the control points are given by

$$C^{(\alpha)} = \begin{bmatrix} 0, & \psi(\theta^{(\alpha)})_{1:n-1}, & 1 \end{bmatrix}, \quad C^{(\sigma)} = \begin{bmatrix} 0, & \psi(\theta^{(\sigma)})_{1:n-1}, & 1 \end{bmatrix}, \tag{12}$$

where $\phi(\theta)_i = \frac{e^{\theta_i}}{\sum_{j=1}^p e^{\theta_j}}$ is a softmax function, and $\psi(\theta)_i = \sum_{j=1}^i \phi(\theta)_j$ is a cumulative softmax function that ensures monotonicity. This monotonic parameterization ensures that $\bar{\rho}(s) = \bar{\alpha}(s)/\bar{\sigma}(s)$ is strictly nondecreasing on $[0,1)$, resulting in $\bar{\rho}^{-1}$ exists.

## 4.4 CONNECTION TO PRIOR WORK

**LD3 (Tong et al., 2025).** From a parameterization perspective, both LD3 and ours optimize the same type of parameter: a nondecreasing sequence of timesteps. However, their interpretations differ: LD3's parameters correspond directly to discrete ODE solver timesteps, whereas ours correspond to Bézier control points that form a continuous sampling path. Interpreting these parameters as an SI scheduler allows our approach to explore a much broader search space compared to LD3. See App. B for the proof.

**Bespoke Solver (Shaul et al., 2024).** Bespoke solver similarly optimizes a new, target sampling path, but the key difference from our approach lies in the parameterization. Their parameterization

is discrete: they learn per-step variables $t_s$ and $c_s$ in Eq. 6. Such a discrete parameterization requires separately modeling the derivatives $\dot{t}_s, \dot{c}_s$, thereby breaking the intrinsic connection between the values and their derivatives. This can yield mismatches between the predicted next value from numerical integration and the actual learned value, ultimately causing unstable optimization.

In contrast, our Bézier parameterization ensures that the resulting scheduler $(\bar{\alpha}_s^\theta, \bar{\sigma}_s^\theta)$ is smooth and, in particular, satisfies the $C^2$ condition required for the SI scheduler as discussed in Sec. 3. Consequently, the time-derivative terms in the transformed velocity can be computed directly rather than learned separately, and the learned ODE trajectories are well-defined, thereby leading to more stable optimization.

Beyond the parameterization, Bespoke Solver also differs in its training objective: it relies on step-wise error minimization, whereas our method optimizes a global trajectory-level loss. See App. C for the results comparing with Bespoke Solver trained under our loss, which isolate and highlight the benefit of our Bézier-based continuous parameterization.

## 5 EXPERIMENTS

**Experiment Setup.** We evaluate our BézierFlow (BF) on both diffusion and flow models across diverse datasets for image generation. For diffusion models, we adopt EDM (Karras et al., 2022) with pretrained checkpoints on CIFAR-10 ($32 \times 32$) (Krizhevsky, 2009), FFHQ ($64 \times 64$) (Karras et al., 2019), and AFHQv2 ($64 \times 64$) (Choi et al., 2020). For flow models, we use pretrained ReFlow (Liu et al., 2023) on CIFAR-10 ($32 \times 32$) (Krizhevsky, 2009), FlowDCN (Wang et al., 2024) on ImageNet ($256 \times 256$) (Deng et al., 2009), and Stable Diffusion v3.5 (SD) (Esser et al., 2024) on MS-COCO ($512 \times 512$) (Lin et al., 2014). All pretrained models are from their official implementations.

Each model is paired with its dedicated ODE solver: UniPC (Zhao et al., 2023) and iPNDM (Zhang & Chen, 2023) for diffusion models, and Runge–Kutta integrators, RK1 (Euler) and RK2 (Midpoint), for flow models. We consider the following learning-based acceleration methods as baselines for comparisons with ours: DMN (Xue et al., 2024), GITS (Chen et al., 2024), and LD3 (Tong et al., 2025). For flow models, we additionally include Bespoke Solver (Shaul et al., 2024) as a baseline, which was specifically designed for RK1 and RK2 solvers. The results of the base ODE solvers, without additional learning, are reported as reference. All baselines are evaluated using their official implementations, except for Bespoke Solver, whose official code is not publicly available.

**Implementation Details.** Methods based on the teacher-forcing framework, including LD3, Bespoke Solver, and our BézierFlow, are trained and validated with the same number of samples for both training and validation: 200 each for CIFAR-10, 50 each for FFHQ, AFHQv2, and ImageNet, and 25 each for Stable Diffusion v3.5, following the experiment setup used in LD3 (Tong et al., 2025). Training is performed for 8 epochs on CIFAR-10, FFHQ, AFHQv2 and 5 epochs on the others. For GITS (Chen et al., 2024), we precompute statistics using 256 sampling trajectories. For the training of BézierFlow, we use 32 control points for the Bézier parameterization unless stated otherwise. For all models, we initialize the target scheduler as the linear SI scheduler, i.e., $\bar{\alpha}(s) = s$ and $\bar{\sigma}(s) = 1 - s$. We set the timesteps uniformly in SNR $\rho(s)$ for diffusion models and uniformly in time $s$ for flow models. Following LD3 (Tong et al., 2025), we also feed the learned decoupled timesteps (Li et al., 2024) to the neural network. See App. D for more details.

**Quantitative Results.** We report Fréchet Inception Distance (FID) (Heusel et al., 2017) scores across diverse datasets for diffusion models in Tab. 1 and for flow models in Tab. 2. FID is computed between the reference set and 50K generated samples, where the test set for each dataset serves as the reference. For SD, both the reference and generated sets are constructed from disjoint subsets of 30K text prompts from MS-COCO, following the setup used in LD3 (Tong et al., 2025). Refer to App. E.2 for a more comprehensive comparison of SD, including text-image alignment metrics.

As shown in Tab. 1, for few-step generation with pretrained diffusion models, BézierFlow consistently achieves the best FID on CIFAR-10 across different NFEs, with especially large margins over the second-best at small NFEs (e.g., at NFE=4, BézierFlow: 9.55 vs. LD3: 12.04 with UniPC, and BézierFlow: 6.93 vs. LD3: 9.97 with iPNDM). On FFHQ and AFHQv2, BézierFlow outperforms or remains comparable to the baselines. The improvements are particularly strong at small NFEs, for example, BézierFlow: 17.05 vs. LD3 (the second-best): 22.48 at NFE=4 with UniPC.

Table 1: **FID comparison of few-step generation with diffusion models**. Results of the base ODE solvers are reported on each top rows, highlighted in gray. **Bold** indicates the best results, and underline marks the second best.

| Method | NFE=4 | NFE=6 | NFE=8 | NFE=10 | Method | NFE=4 | NFE=6 | NFE=8 | NFE=10 |
|---|---|---|---|---|---|---|---|---|---|
| CIFAR-10 $32 \times 32$ with EDM (Karras et al., 2022) (Teacher FID: 2.08) | | | | | | | | | |
| UniPC | 50.30 | 19.33 | 9.64 | 6.16 | iPNDM | 29.53 | 9.84 | 5.30 | 3.75 |
| + DMN | 26.42 | 8.11 | 4.22 | 2.79 | + DMN | 28.29 | 9.33 | 4.82 | 3.52 |
| + GITS | 24.83 | 11.02 | 6.68 | 5.02 | + GITS | 16.20 | 6.80 | 4.07 | 3.30 |
| + LD3 | _12.04_ | _3.56_ | _2.43_ | _2.62_ | + LD3 | 9.97 | _4.42_ | _2.93_ | _2.44_ |
| + BézierFlow | **9.55** | **3.13** | **2.40** | **2.09** | + BézierFlow | **6.93** | **3.35** | **2.81** | **2.43** |
| FFHQ $64 \times 64$ with EDM (Karras et al., 2022) (Teacher FID: 2.86) | | | | | | | | | |
| UniPC | 47.62 | 14.96 | 7.76 | 8.93 | iPNDM | 28.75 | 11.15 | 6.68 | 4.80 |
| + DMN | 25.87 | 9.44 | 5.06 | 4.06 | + DMN | 30.89 | 11.93 | 7.33 | 6.20 |
| + GITS | 22.99 | 12.12 | 8.90 | 4.40 | + GITS | 18.51 | 9.21 | 5.58 | 4.37 |
| + LD3 | _22.48_ | **6.16** | _4.25_ | **2.92** | + LD3 | _15.55_ | **5.89** | **3.74** | **3.03** |
| + BézierFlow | **17.05** | _7.43_ | **3.82** | _3.13_ | + BézierFlow | **15.39** | _7.84_ | _5.56_ | _3.75_ |
| AFHQv2 $64 \times 64$ with EDM (Karras et al., 2022) (Teacher FID: 2.04) | | | | | | | | | |
| UniPC | 23.59 | 10.15 | 7.76 | 6.38 | iPNDM | 15.14 | 6.12 | 3.80 | 3.01 |
| + DMN | 30.39 | 14.40 | 3.98 | 3.69 | + DMN | 33.21 | 15.95 | 5.99 | 5.29 |
| + GITS | _13.20_ | 7.50 | 3.89 | 3.94 | + GITS | 14.31 | 5.81 | 3.88 | 3.57 |
| + LD3 | 18.17 | _4.95_ | **2.68** | _3.02_ | + LD3 | **11.85** | 3.11 | 2.45 | _2.18_ |
| + BézierFlow | **12.27** | **4.46** | _2.75_ | **2.67** | + BézierFlow | _14.44_ | 4.69 | 2.63 | **2.16** |

Table 2: **FID comparison of few-step generation with flow-based models**. Results of the base ODE solvers are reported on each top rows, highlighted in gray. **Bold** indicates the best results, and underline marks the second best.

| Method | NFE=4 | NFE=6 | NFE=8 | NFE=10 | Method | NFE=4 | NFE=6 | NFE=8 | NFE=10 |
|---|---|---|---|---|---|---|---|---|---|
| CIFAR-10 $32 \times 32$ with ReFlow (Liu et al., 2023) (Teacher FID: 2.70) | | | | | | | | | |
| RK1 | 52.78 | 26.30 | 17.40 | 13.30 | RK2 | 25.36 | 12.12 | 9.17 | 7.89 |
| + DMN | 180.03 | 104.23 | 30.94 | 21.58 | + DMN | 82.41 | 51.99 | 21.43 | 18.62 |
| + Bespoke | 45.31 | 18.08 | 11.88 | 9.25 | + Bespoke | 39.45 | 64.87 | 16.67 | 13.34 |
| + GITS | 47.42 | 26.11 | 19.89 | 15.34 | + GITS | _22.84_ | _11.84_ | 8.77 | 6.58 |
| + LD3 | _38.95_ | _20.10_ | _12.54_ | _9.64_ | + LD3 | 29.45 | 13.82 | _6.26_ | _3.86_ |
| + BézierFlow | **20.64** | **9.67** | **7.30** | **5.51** | + BézierFlow | **13.18** | **6.00** | **4.31** | **3.74** |
| ImageNet $256 \times 256$ with FlowDCN (Wang et al., 2024) (Teacher FID: 15.89) | | | | | | | | | |
| RK1 | 12.03 | 12.04 | 13.55 | 14.43 | RK2 | 7.91 | 10.54 | 12.97 | 14.08 |
| + DMN | 142.79 | 28.56 | _10.61_ | _11.69_ | + DMN | 7.96 | 10.23 | _9.42_ | _7.86_ |
| + Bespoke | _11.85_ | 11.81 | 13.39 | 14.31 | + Bespoke | _7.66_ | 10.05 | 13.02 | 14.23 |
| + GITS | 13.20 | _10.91_ | 11.91 | 12.93 | + GITS | 8.18 | _9.80_ | 12.30 | 13.27 |
| + LD3 | **11.62** | 11.94 | 13.36 | 14.12 | + LD3 | 7.59 | 10.17 | 12.75 | 14.04 |
| + BézierFlow | 15.60 | **6.85** | **7.77** | **8.11** | + BézierFlow | 9.50 | **5.94** | **6.22** | **7.56** |
| MS-COCO $512 \times 512$ with Stable Diffusion (Esser et al., 2024) (Teacher FID: 12.13) | | | | | | | | | |
| RK1 | 57.93 | **30.96** | 21.50 | _17.19_ | RK2 | 34.95 | 17.89 | 13.33 | 11.61 |
| + DMN | 113.24 | 46.02 | 31.58 | 24.41 | + DMN | 36.33 | _16.45_ | 27.09 | 17.36 |
| + Bespoke | 134.21 | 52.51 | 23.70 | 20.69 | + Bespoke | 45.23 | 40.87 | 20.18 | 13.26 |
| + GITS | 70.01 | 42.44 | 31.89 | 25.47 | + GITS | **31.09** | 21.21 | 15.58 | 14.65 |
| + LD3 | _55.31_ | 36.85 | _20.37_ | 19.76 | + LD3 | 39.03 | 18.04 | _12.30_ | _11.54_ |
| + BézierFlow | **54.05** | _33.43_ | **19.69** | **16.52** | + BézierFlow | _33.94_ | **16.41** | **12.20** | **11.02** |

When it comes to flow models, as shown in Tab. 2, BézierFlow achieves state-of-the-art results on CIFAR-10 with both RK1 and RK2, outperforming the others by clear margins. For example, we surpass the second-best LD3 by 18.31 at NFE=4 with RK1, and the second-best GITS by 9.66 at NFE=4 with RK2. On ImageNet, we consistently obtain the best results across most NFEs, except at NFE=4, again by large margins. On MS-COCO evaluated with Stable Diffusion v3.5, BézierFlow outperforms the baselines at most NFEs, demonstrating generalizability to large-scale models.

Overall, these results demonstrate that BézierFlow attains the best or comparable performance to existing acceleration approaches across diverse experiment setups, including both diffusion and flow models, different NFEs, ODE solvers, and datasets.

**Qualitative Results.** We present qualitative results for accelerated sampling of diffusion models in Fig. 2 and flow models in Fig. 3. Across both model classes, BézierFlow (BF) consistently produces

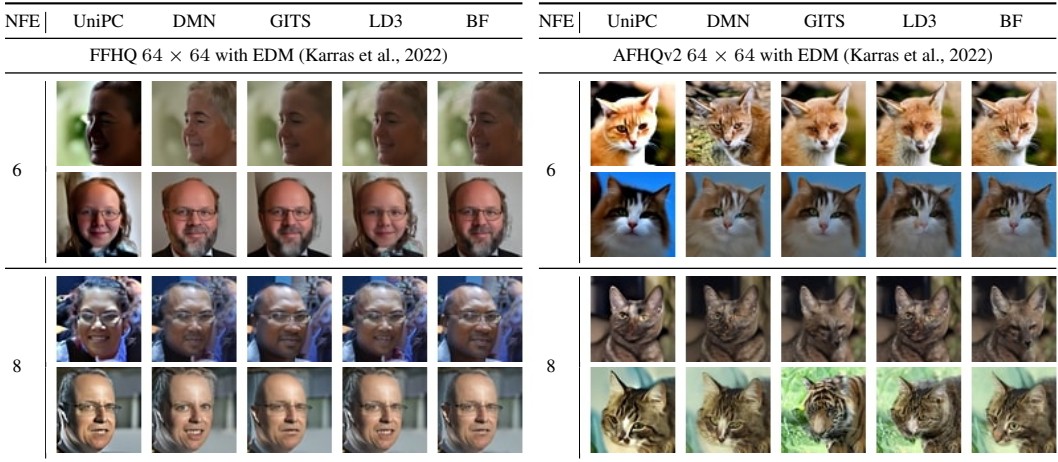

Figure 2: **Qualitative comparisons of samples generated using NFEs 6 and 8 on FFHQ and AFHQv2 datasets.** We use UniPC solver as the base solver for both cases.

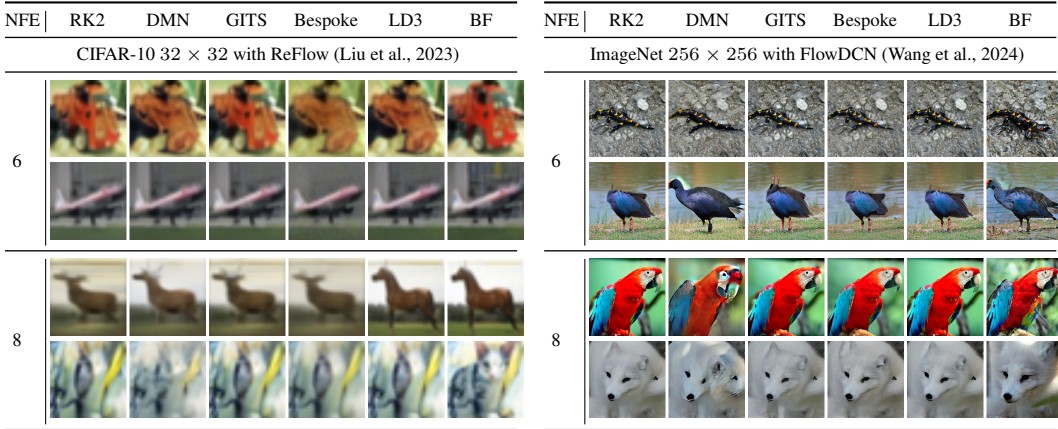

Figure 3: **Qualitative comparisons of samples generated using NFEs 6 and 8 on CIFAR-10 and ImageNet datasets.** We use RK2 solver as the base solver for both cases.

sharper details and fewer artifacts at low NFEs. Notably, in the last row of Fig. 2 (right), the baselines fail to generate a plausible animal face, whereas our method produces a clear and realistic cat face. See App. F for more qualitative results.

Table 3: **Generalizability of BézierFlow to unseen NFEs.** Each column corresponds to the inference NFE. Baselines are trained with the same NFE used at inference, whereas ours is trained once with NFE=10 and directly applied to unseen NFEs. Reported results are FID scores on CIFAR-10 (lower is better; best in **bold**).

| Method | RK1 | | | RK2 | | |
|---|---|---|---|---|---|---|
| | 6 | 8 | 10 | 6 | 8 | 10 |
| GITS | 26.11 | 19.89 | 15.34 | 11.84 | 8.77 | 6.58 |
| Bespoke | 18.08 | 11.88 | 9.25 | 64.87 | 16.67 | 13.34 |
| LD3 | 20.10 | 12.54 | 9.64 | 13.82 | 6.26 | 3.86 |
| BF (NFE=10) | **18.50** | **9.02** | **6.01** | **9.57** | **5.32** | **3.71** |

Table 4: **Comparison with distillation methods on CIFAR-10.** The middle column reports FID and NFE. Our training time is measured on an NVIDIA A6000, while distillation times are directly from the original papers on NVIDIA A100s. Best in **bold**.

| Method | FID (↓) / NFE | Training Time |
|---|---|---|
| CIFAR-10 32×32 — Diffusion | | |
| CD | 2.93 / NFE=2 | 8 days |
| BF w/ UniPC | **2.09** / NFE=10 | 15 minutes |
| CIFAR-10 32×32 — Flow | | |
| 2-RF | 3.85 / NFE=2 | 8 days |
| BF w/ RK2 | **3.74** / NFE=10 | 15 minutes |

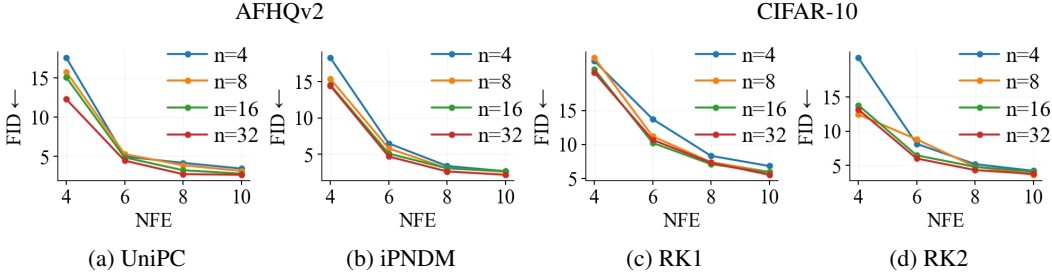

Figure 4: **Effect of the degree of Bézier functions.** Each line reports FID scores on CIFAR-10 across Bézier function degrees $n$ from 4 to 32. Higher degrees yield lower FID, with convergence around $n = 32$.

**Effect of the Degree of Bézier Functions.** Fig. 4 shows the effect of the degree of Bézier functions by varying the number of control points from 4 to 32. Across different datasets, NFEs, and base ODE solvers, increasing the number of control points consistently improves FID, indicating that higher-degree Bézier functions provide greater expressiveness for learning optimal sampling trajectories. Empirically, we observe that the additional gains between $n = 16$ and $n = 32$ become marginal, so we adopt $n = 32$ as our default. Note that the training time overhead from increasing $n$ is negligible, with only a few seconds difference between $n = 4$ and $n = 32$.

**Generalizability to Unseen NFEs.** As discussed in Sec. 4.4, unlike prior works (Xue et al., 2024; Chen et al., 2024; Tong et al., 2025; Shaul et al., 2024) that learn *discrete per-step variables*, BézierFlow learns the sampling trajectory with continuous functions, enabling generalization to NFEs unseen during training. As shown in Tab. 3, BézierFlow trained with NFE=10 also performs well at NFE=6 and NFE=8, achieving outperforming FID scores against the baselines trained directly at those NFEs.

**Training Efficiency.** In Tab. 4, we compare BézierFlow on CIFAR-10 (Krizhevsky, 2009) against representative distillation-based approaches: Consistency Distillation (CD) (Song et al., 2023) for diffusion models and 2-Rectified Flow (2-RF) (Liu et al., 2023) for flow models. While BézierFlow achieves better FID scores with a larger inference NFE, its *training* cost is significantly lower, requiring only **15 minutes** compared to **8 days** for distillation, which corresponds to approximately **0.13%** of the training time. A more comprehensive comparison of training time is provided in App. E.3.

**Combination with LD3.** Although the search space of LD3 is a subset of that of BézierFlow, LD3 optimizes discrete timesteps whereas BézierFlow learns continuous sampling paths, making it natural to ask whether jointly optimizing the two could offer complementary improvements. We therefore optimize both the target timesteps and the scheduler in a unified framework. As shown in Tab. 5 of the Appendix, however, the combination does not offer clear advantages over using BézierFlow alone. This indicates that the advantages provided by LD3 are already captured within the learned scheduler, and thus do not translate into further gains when optimized jointly.

## 6 CONCLUSION

We introduce BézierFlow, a lightweight training framework for few-step generation. By combining the optimization of sampling trajectories, rather than discrete ODE timesteps, with a Bézier-based continuous parameterization, BézierFlow achieves consistent improvements across diffusion and flow models with only minutes of training, surpassing existing lightweight training approaches. For future work, we plan to explore alternative basis functions for Bézier functions, which may enable richer expressiveness with fewer control points.

**Ethics Statement.** We affirm adherence to the ICLR Code of Ethics. This work relies only on publicly available models and datasets and does not involve human subjects, user data, or personally identifiable information. We acknowledge the potential for misuse of generative AI and encourage responsible deployment and use of our method.

## ACKNOWLEDGEMENTS

This work was supported by IITP grants (RS-2024-00399817, RS-2025-25441313, RS-2025-25443318), funded by the Korean government (MSIT); the Industrial Technology Innovation Program (RS-2025-02317326), funded by the Korean government (MOTIE); the National Supercomputing Center (KSC-2025-CRE-0475); and the DRB-KAIST SketchTheFuture Research Center.

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

## A  VALIDITY OF SCHEDULER REPARAMETERIZATION

We provide a detailed exposition of two key properties discussed in Sec. 4.2: (i) the preservation of endpoint marginals, and (ii) the invaraince of the SI training objective with respect to the choice of schedule, provided that the SNR endpoints are identical.

**Proposition A.1.** *(Endpoint marginals equivalence) Let $p_0$ and $p_1$ denote the endpoint marginals of $x_t$ governed by a scheduler $(\alpha_t, \sigma_t)$, $\bar{p}_0$ and $\bar{p}_1$ denote those of another stochastic interpolant $\bar{x}_s$. If $\bar{x}_s = c_s x_{t_s}$ with $c_s$ and $t_s$ defined in Eq. 6, then the endpoint marginals are preserved, i.e., $p_0 = \bar{p}_0$ and $p_1 = \bar{p}_1$.*

*Proof.* By the boundary conditions discussed in Sec. 3, any SI scheduler satisfies

$$\alpha(0) = 0, \ \alpha(1) = 1, \ \sigma(0) = 1, \ \sigma(1) = 0. \tag{13}$$

From Eq. 6, $c_s = 1$ at both $s = 0$ and $s = 1$. By the definition of $t_s = \rho^{-1}(\bar{\rho}(s))$, we also have $t_{s=0} = 0$ and $t_{s=1} = 1$. Hence, $\bar{x}_{s=0} = x_{t=0}$ and $\bar{x}_{s=1} = x_{t=1}$. Since the pair of endpoints $(x_0, x_1)$ does not change under the sampling path transformation, the endpoint marginals also coincide: $p_0 = \bar{p}_0$ and $p_1 = \bar{p}_1$. $\square$

**Proposition A.2.** *(Training objective invariance) For any two schedulers $(\alpha_t, \sigma_t)$ and $(\bar{\alpha}_s, \bar{\sigma}_s)$ with matching SNR endpoints, training an SI model $S_\phi$ under either scheduler minimizes the same training objective, hence yields equivalent optima.*

*Proof.* As noted in Sec. 3, different SI models (noise, velocity, or score predictors) learn different but interchangeable quantities, and can all be expressed in the denoiser form $\hat{x}_\phi$. For convenience, we therefore recall the continuous-time SI objective in the denoiser form (Eq. 18 from Kingma et al. (2023)): for a schedule $(\alpha_t, \sigma_t)$ with $\rho(t) = \alpha(t)/\sigma(t)$ strictly increasing, define $\nu = \rho^2$ and

$$x_\nu = \alpha_\nu x_1 + \sigma_\nu x_0, \qquad \alpha_\nu := \alpha\big(\rho^{-1}(\sqrt{\nu})\big), \ \ \sigma_\nu := \sigma\big(\rho^{-1}(\sqrt{\nu})\big), \tag{14}$$

with $\nu_{\min} = \rho(0)^2$, $\nu_{\max} = \rho(1)^2$. Then,

$$\mathcal{L}_\nu[\phi; (\alpha_t, \sigma_t)] = \frac{1}{2} \int_{\nu_{\min}}^{\nu_{\max}} \mathbb{E}_{x_1 \sim p_{\mathrm{data}}, \, x_0 \sim p_0}\big[\|x_1 - \hat{x}_\phi(x_\nu, \nu)\|_2^2\big] \ d\nu. \tag{15}$$

Now consider another schedule $(\bar{\alpha}_s, \bar{\sigma}_s)$ with the same SNR endpoints. Since $\rho(t) = \frac{\alpha(t)}{\sigma(t)}$ and $\bar{\rho}(s) = \frac{\bar{\alpha}(s)}{\bar{\sigma}(s)}$ are strictly increasing, the maps $t \mapsto \nu = \rho(t)^2$ and $s \mapsto \nu = \bar{\rho}(s)^2$ are bijections onto the common interval $[\nu_{\min}, \nu_{\max}]$. Now, for each fixed $\nu$, we have

$$\nu = \frac{\alpha(t)^2}{\sigma(t)^2} = \frac{\bar{\alpha}(s)^2}{\bar{\sigma}(s)^2}, \tag{16}$$

which implies $\sigma_\nu = \frac{\alpha_\nu}{\sqrt{\nu}}$ and $\bar{\sigma}_\nu = \frac{\bar{\alpha}_\nu}{\sqrt{\nu}}$. Therefore, the interpolants satisfy

$$x_\nu = \alpha_\nu\Big(x_1 + \frac{1}{\sqrt{\nu}}x_0\Big), \quad \bar{x}_\nu = \bar{\alpha}_\nu\Big(x_1 + \frac{1}{\sqrt{\nu}}x_0\Big), \tag{17}$$

so that $x_\nu = \frac{\alpha_\nu}{\bar{\alpha}_\nu} \bar{x}_\nu$. This shows that the two interpolants differ only by a scalar rescaling factor, and since the integration limits are the same, we conclude

$$\mathcal{L}_\nu[\phi; (\alpha_t, \sigma_t)] = \mathcal{L}_\nu[\phi; (\bar{\alpha}_s, \bar{\sigma}_s)]. \tag{18}$$

$\square$

## B  THEORETICAL ANALYSIS OF SAMPLING TRAJECTORY SPACES

In this section, we formally show that the family of sampling trajectories realizable by BézierFlow is a super set of that of LD3 (Tong et al., 2025), offering better optimization advantages.

**Theorem B.1** (Inclusion of LD3 in the BézierFlow Trajectory Space). *Let $M$ and $D$ denote the number of sampling steps and the dimension of the state $x$, respectively. Let $\mathcal{X}_{\mathrm{LD3}}, \mathcal{X}_{\mathrm{BF}} \subseteq \mathbb{R}^{M+1 \times D}$ be the sets of sampling trajectories over discrete timesteps realizable by LD3 and BézierFlow (parameterized by Bézier curves of degree $n \geq M$). Assuming the source sampling path defines a non-linear geometry, the trajectory space of LD3 is a subset of that of BézierFlow:*

$$\mathcal{X}_{\mathrm{LD3}} \subsetneq \mathcal{X}_{\mathrm{BF}}. \tag{19}$$

*Proof.* Let $\{t_k\}_{k=0}^M$ and $\{s_k\}_{k=0}^M$ denote the timesteps on the source scheduler and on the trajectory induced by the Bézier SI scheduler, respectively, with

$$0 = t_0 < t_1 < \cdots < t_M = 1, \qquad 0 = s_0 < s_1 < \cdots < s_M = 1.$$

Define

$$\alpha_k := \alpha(t_k), \quad \sigma_k := \sigma(t_k), \quad x_{t_k} = \alpha_k x_1 + \sigma_k x_0, \qquad k = 0, \ldots, M. \tag{20}$$

With this notation, any LD3 sampling trajectory over $M$ timesteps can be written as $\mathbf{x} = (x_{t_0}, \ldots, x_{t_M}) \in \mathcal{X}_{\mathrm{LD3}}$.

Since a Bézier curve of degree $n \geq M$ can interpolate any $M + 1$ distinct values, there exists $\theta^\star$ such that

$$\bar{\alpha}_{\theta^\star}(s_k) = \alpha_k, \qquad \bar{\sigma}_{\theta^\star}(s_k) = \sigma_k, \qquad \forall k. \tag{21}$$

Hence

$$\bar{\rho}_{\theta^\star}(s_k) := \frac{\bar{\alpha}_{\theta^\star}(s_k)}{\bar{\sigma}_{\theta^\star}(s_k)} = \frac{\alpha_k}{\sigma_k} = \rho(t_k), \tag{22}$$

and therefore

$$t_{s_k} = \rho^{-1}\big(\bar{\rho}_{\theta^\star}(s_k)\big) = t_k, \qquad c_{s_k} = \frac{\bar{\sigma}_{\theta^\star}(s_k)}{\sigma(t_{s_k})} = \frac{\sigma_k}{\sigma_k} = 1. \tag{23}$$

Using the sampling transformation in Eq. 8,

$$\bar{x}_{s_k} = c_{s_k} x_{t_{s_k}} = x_{t_k}, \qquad \forall k, \tag{24}$$

so every $\mathbf{x} \in \mathcal{X}_{\mathrm{LD3}}$ is also realizable by BézierFlow, and thus

$$\mathcal{X}_{\mathrm{LD3}} \subseteq \mathcal{X}_{\mathrm{BF}}. \tag{25}$$

For strictness, fix $\{t_k\}_{k=0}^M$ and consider a target scheduler $\theta$ such that

$$\bar{\rho}_\theta(s_k) = \rho(t_k), \qquad \bar{\sigma}_\theta(s_k) \neq \sigma_k \quad \text{for at least one } k. \tag{26}$$

Then

$$t_{s_k} = \rho^{-1}\big(\bar{\rho}_\theta(s_k)\big) = t_k, \qquad c_{s_k} = \frac{\bar{\sigma}_\theta(s_k)}{\sigma(t_{s_k})} = \frac{\bar{\sigma}_\theta(s_k)}{\sigma_k} \neq 1 \quad \text{for at least one } k, \tag{27}$$

and the resulting trajectory satisfies

$$\bar{x}_{s_k} = c_{s_k} x_{t_k}, \qquad c_{s_k} \neq 1 \quad \text{for some } k. \tag{28}$$

Since LD3 is constrained to the fixed source scheduler, which corresponds to sampling via Eq. 8 with

$$s_k = t_k, \qquad c_{s_k} \equiv 1, \tag{29}$$

so any trajectory with $c_{s_k} \neq 1$ for some $k$ cannot be realized by LD3. Thus

$$\mathbf{x}_\theta \in \mathcal{X}_{\mathrm{BF}} \quad \text{and} \quad \mathbf{x}_\theta \notin \mathcal{X}_{\mathrm{LD3}}, \tag{30}$$

which implies

$$\mathcal{X}_{\mathrm{LD3}} \subsetneq \mathcal{X}_{\mathrm{BF}}. \tag{31}$$

$\square$

Table 5: **FID comparison of BézierFlow, LD3 and their combination, denoted as Both.** The best results are highlighted in **bold** and the second best results are underlined. Gray cells indicate the base ODE solvers.

| Method | NFE=4 | NFE=6 | NFE=8 | NFE=10 | Method | NFE=4 | NFE=6 | NFE=8 | NFE=10 |
|---|---|---|---|---|---|---|---|---|---|
| UniPC | CIFAR-10 with EDM (Teacher FID: 2.08) | | | | UniPC | FFHQ with EDM (Teacher FID: 2.86) | | | |
| + LD3 | 12.04 | 3.56 | 2.43 | 2.62 | + LD3 | 22.48 | 6.16 | 4.25 | 2.92 |
| + BézierFlow | 9.55 | **3.13** | **2.40** | **2.09** | + BézierFlow | **17.05** | 7.43 | **3.82** | 3.13 |
| + Both | **9.32** | 3.37 | 2.44 | 2.71 | + Both | 20.77 | 6.24 | 4.13 | **3.04** |
| RK2 | CIFAR-10 with ReFlow (Teacher FID: 2.70) | | | | RK2 | ImageNet with FlowDCN (Teacher FID: 15.89) | | | |
| + LD3 | 29.45 | 13.82 | 6.26 | 3.86 | + LD3 | **7.59** | 10.17 | 12.75 | 14.04 |
| + BézierFlow | 13.18 | 6.00 | 4.31 | 3.74 | + BézierFlow | 9.50 | **5.94** | **6.22** | **7.56** |
| + Both | **12.23** | **5.50** | **3.74** | **3.17** | + Both | 9.11 | 6.64 | 9.38 | 10.88 |

Table 6: **FID comparison of BézierFlow, Bespoke Solver and Bespoke Solver trained with our training loss, denoted as Bespoke\*.** Results for the base solvers are reported on each top rows. The best results are highlighted in **bold** and the second best results are underlined. Gray cells indicate the base ODE solvers.

| Method | NFE=4 | NFE=6 | NFE=8 | NFE=10 | Method | NFE=4 | NFE=6 | NFE=8 | NFE=10 |
|---|---|---|---|---|---|---|---|---|---|
| | CIFAR-10 32 × 32 with ReFlow (Liu et al., 2023) (Teacher FID: 2.70) | | | | | | | | |
| RK1 | 52.78 | 26.30 | 17.40 | 13.30 | RK2 | 25.36 | 12.12 | 9.17 | 7.89 |
| + Bespoke | 45.31 | 18.08 | 11.88 | 9.25 | + Bespoke | 39.45 | 64.87 | 16.67 | 13.34 |
| + Bespoke\* | 38.34 | 17.28 | 10.34 | 7.65 | + Bespoke\* | 19.44 | 49.65 | 4.40 | **3.70** |
| + BézierFlow | **20.64** | **9.67** | **7.30** | **5.51** | + BézierFlow | **13.18** | **6.00** | **4.31** | 3.74 |

**Proposition B.2** (Better Optima under Larger Trajectory Spaces). *Let $\mathcal{X}_{\text{LD3}}, \mathcal{X}_{\text{BF}} \subseteq \mathbb{R}^{M+1 \times D}$ be the trajectory spaces of LD3 and BézierFlow, respectively, and suppose*

$$\mathcal{X}_{\text{LD3}} \subsetneq \mathcal{X}_{\text{BF}} \tag{32}$$

*as in Theorem B.1. Let $\mathcal{L} : \mathcal{X} \to \mathbb{R}$ be any real-valued objective functional (e.g., a distillation loss to a teacher). Define the optimal objective values*

$$\mathcal{L}_{\text{LD3}}^{\star} := \inf_{\mathbf{x} \in \mathcal{X}_{\text{LD3}}} \mathcal{L}(\mathbf{x}), \qquad \mathcal{L}_{\text{BF}}^{\star} := \inf_{\mathbf{x} \in \mathcal{X}_{\text{BF}}} \mathcal{L}(\mathbf{x}). \tag{33}$$

*Then, the following inequality holds:*

$$\mathcal{L}_{\text{BF}}^{\star} \leq \mathcal{L}_{\text{LD3}}^{\star}. \tag{34}$$

*Proof.* Recall that for any two sets $\mathcal{A} \subseteq \mathcal{B}$ and an objective function $f$, the infimum over the superset is less than or equal to the infimum over the subset, i.e.,

$$\inf_{x \in \mathcal{B}} f(x) \leq \inf_{x \in \mathcal{A}} f(x). \tag{35}$$

Since $\mathcal{X}_{\text{LD3}} \subsetneq \mathcal{X}_{\text{BF}}$, applying this property directly yields:

$$\mathcal{L}_{\text{BF}}^{\star} = \inf_{\mathbf{x} \in \mathcal{X}_{\text{BF}}} \mathcal{L}(\mathbf{x}) \leq \inf_{\mathbf{x} \in \mathcal{X}_{\text{LD3}}} \mathcal{L}(\mathbf{x}) = \mathcal{L}_{\text{LD3}}^{\star}. \tag{36}$$

Moreover, if there exists $\mathbf{x}' \in \mathcal{X}_{\text{BF}} \setminus \mathcal{X}_{\text{LD3}}$ such that $\mathcal{L}(\mathbf{x}') < \mathcal{L}_{\text{LD3}}^{\star}$, then

$$\mathcal{L}_{\text{BF}}^{\star} < \mathcal{L}_{\text{LD3}}^{\star}. \tag{37}$$

□

## C   COMPARISON OF SCHEDULER PARAMETERIZATIONS: BÉZIERFLOW VS. BESPOKE SOLVER

As discussed in Sec.4.4, both Bespoke Solver (Shaul et al., 2024) and BézierFlow aim to learn sampling trajectories, but differ in (i) parameterization and (ii) training objective. Bespoke Solver (Shaul et al., 2024) employs discrete per-step parameterization and minimizes step-wise $\ell_2$ errors against

teacher outputs, whereas BézierFlow adopts a Bézier-based continuous parameterization and is trained with a global truncation loss, computed along the full trajectory from $x_0$ to $x_1$, with LPIPS (Zhang et al., 2018).

To ablate the effect of different training objectives and focus solely on parameterization, we report additional quantitative results in Tab. 6, where Bespoke* retains Bespoke Solver's parameterization but adopts the same training objective as ours. While Bespoke* improves over the original Bespoke Solver, BézierFlow remains superior, with especially large gains at low NFEs, such as +17.7 FID improvement at NFE=4 with RK1, underscoring the advantage of our Bézier-based continuous parameterization.

## D   IMPLEMENTATION DETAILS

We first describe the shared experimental setup for the methods based on teacher-forcing framework (Bespoke solver, LD3, and BézierFlow) and method-specific configurations for BézierFlow and the baselines.

### D.1   SHARED SETUP

**Teacher Data Generation.**   We generate teacher samples using the high-order adaptive solver RK45 (Butcher, 1996), except for Stable Diffusion v3.5 (Esser et al., 2024), where we adopt RK2 with 30 NFEs. The same teacher samples are used for all baselines that rely on the teacher-forcing framework (e.g., Bespoke Solver, LD3).

**Training.**   We train for 8 epochs on CIFAR-10 (Krizhevsky, 2009), FFHQ (Karras et al., 2019), and AFHQv2 (Choi et al., 2020), and for 5 epochs on ImageNet (Deng et al., 2009) and Stable Diffusion v3.5 (Esser et al., 2024). At the end of each epoch, we perform validation and select the checkpoint with the best validation score for final evaluation. We use LPIPS (Zhang et al., 2018) as the distance metric for LD3 (Tong et al., 2025) and BézierFlow, and RMSE for Bespoke Solver (Shaul et al., 2024).

**Evaluation.**   We report Fréchet Inception Distance (FID) (Heusel et al., 2017) scores computed against the reference set using 50K randomly generated samples. On ImageNet, generated samples are drawn to match the class distribution of the reference set. For SD3.5, both reference and generated samples are constructed from disjoint subsets of 30K text prompts from the MS-COCO validation set, following the setup of LD3 (Tong et al., 2025).

### D.2   BÉZIERFLOW TRAINING DETAILS

**Target Timesteps.**   For diffusion models, the timesteps $\{s_i\}_{i=0}^{\text{NFE}}$ are initialized to be uniformly spaced in terms of the signal-to-noise ratio (SNR). For flow models, they are initialized to be uniformly spaced in the time domain.

**Initialization.**   We initialize the Bézier scheduler with a linear SI scheduler, i.e., $\bar{\alpha}(s) = s$ and $\bar{\sigma}(s) = 1 - s$. Under the 1D Bézier parameterization, this corresponds to

$$\theta_i^{(\alpha)} = 1, \quad \theta_i^{(\sigma)} = 1, \qquad i = 0, 1, \dots, n, \tag{38}$$

which places the $n - 1$ interior control points uniformly between the two endpoints. We use 32 control points in all experiments. For decoupled timesteps $s_i^c$ that are fed into the model, we set $s_i^c = s_i + \theta_i^{(c)}$, with $\theta_i^{(c)}$ initialized to zero, following LD3 (Tong et al., 2025).

**Optimizer.**   We optimize the Bézier scheduler parameters $\theta^{(\alpha)}, \theta^{(\sigma)}$ using RMSprop, and the decoupled timesteps $\theta_i^{(c)}$ using SGD. For RMSprop, we set the momentum to 0.9 and weight decay to 0. The learning rate is $5 \times 10^{-3}$ for CIFAR-10, FFHQ, and AFHQv2, and $1 \times 10^{-3}$ for ImageNet and Stable Diffusion v3.5. For the decoupled timesteps, we use SGD with a learning rate of $1 \times 10^{-1}$ for CIFAR-10 and ImageNet, $1 \times 10^{-2}$ for FFHQ and AFHQv2, and $5 \times 10^{-4}$ for Stable Diffusion v3.5, each further scaled by $1/\text{NFE}$. We apply gradient clipping with a global norm threshold of 1.0 to all parameters.

Table 7: **FID comparison of few-step generation with diffusion models at extremely low NFEs.** Results of the base ODE solvers are reported on each top rows. **Bold** indicates the best results, and underline marks the second best. Gray cells indicate the base ODE solvers.

| Method | CIFAR-10 32 × 32 with EDM | | | FFHQ 64 × 64 with EDM | | | AFHQv2 64 × 64 with EDM | | |
|---|---|---|---|---|---|---|---|---|---|
| | NFE=1 | NFE=2 | NFE=3 | NFE=1 | NFE=2 | NFE=3 | NFE=1 | NFE=2 | NFE=3 |
| UniPC | 377.15 | 168.35 | 57.45 | 280.61 | 104.57 | 59.54 | 312.37 | 64.62 | 44.52 |
| + DMN | - | 160.65 | 66.03 | - | 142.57 | 64.99 | - | 141.99 | 70.01 |
| + GITS | - | 168.29 | 53.21 | - | 107.71 | 42.38 | - | 73.95 | **25.13** |
| + LD3 | - | 187.42 | **39.56** | - | 120.87 | 48.30 | - | 107.77 | 30.53 |
| + BézierFlow | **125.03** | **50.41** | 55.07 | **121.94** | 72.03 | **33.72** | **159.58** | **39.86** | 26.31 |
| iPNDM | 377.15 | 153.31 | 47.68 | 280.61 | 102.50 | 45.70 | 312.37 | 79.32 | 38.16 |
| + DMN | - | 146.40 | 58.98 | - | 112.55 | 61.54 | - | 128.36 | 76.28 |
| + GITS | - | 153.33 | 43.71 | - | 105.78 | **32.33** | - | 95.47 | 26.40 |
| + LD3 | - | 145.03 | 32.19 | - | 97.62 | 38.14 | - | 91.10 | **23.85** |
| + BézierFlow | **125.03** | **41.58** | **22.20** | **121.94** | **60.45** | 35.10 | **159.58** | **34.70** | 36.26 |

Table 8: **FID comparison of few-step generation with flow models at extremely low NFEs.** Results of the base ODE solvers are reported on each top rows. **Bold** indicates the best results, and underline marks the second best. Gray cells indicate the base ODE solvers.

| Method | CIFAR-10 32 × 32 with ReFlow | | | ImageNet 256 × 256 with FlowDCN | | | MS-COCO 512 × 512 with SDv3.5 | | |
|---|---|---|---|---|---|---|---|---|---|
| | NFE=1 | NFE=2 | NFE=3 | NFE=1 | NFE=2 | NFE=3 | NFE=1 | NFE=2 | NFE=3 |
| RK1 | 379.22 | 171.48 | 89.00 | 263.54 | 113.27 | **27.69** | 328.02 | 214.03 | 103.97 |
| + DMN | - | 170.53 | 79.54 | - | 115.89 | 42.78 | - | 218.20 | **82.68** |
| + GITS | - | 183.40 | 81.46 | - | 130.81 | 31.70 | - | 166.06 | 94.94 |
| + Bespoke | 471.18 | 405.94 | 265.77 | 264.81 | 114.66 | 28.27 | 324.94 | 212.16 | 98.91 |
| + LD3 | - | 182.40 | 81.35 | - | 126.29 | 85.16 | - | **150.21** | 85.16 |
| + BézierFlow | **314.52** | **67.63** | **30.40** | **261.79** | 94.64 | 44.60 | **320.67** | 156.20 | 83.55 |
| RK2 | - | 128.80 | - | - | 90.26 | - | - | 163.35 | - |
| + DMN | - | - | - | - | - | - | - | - | - |
| + GITS | - | - | - | - | - | - | - | - | - |
| + Bespoke | - | 309.60 | - | - | 86.58 | - | - | 162.40 | - |
| + LD3 | - | - | - | - | - | - | - | - | - |
| + BézierFlow | - | **70.87** | - | - | **83.97** | - | - | **146.20** | - |

## D.3 BASELINES

**GITS (Chen et al., 2024).** We adopt the official implementation code and follow the default number of sampling trajectories, which is 256.

**Bespoke Solver (Shaul et al., 2024).** Since no official implementation code is publicly available, we re-implemented the method based on the descriptions in the original paper. We employ Adam optimizer with a learning rate of $1 \times 10^{-4}$, as we observed that the learning rate reported in the paper ($2 \times 10^{-3}$) caused divergence and very high FID scores when training on relatively small datasets.

**LD3 (Tong et al., 2025).** We adopt the official implementation code and follow the default training configurations. For timestep parameters, we use the same optimizer and match their learning rate to that of our scheduler. For the decoupled timesteps, we follow the original parameterization and use SGD with a learning rate of $\frac{0.1}{\text{NFE}}$ except for ImageNet and Stable Diffusion v3.5, where we use $\frac{0.001}{\text{NFE}}$ following original paper.

## E MORE QUANTITATIVE RESULTS

### E.1 PROBING BÉZIERFLOW AT EXTREMELY LOW NFES

To stress-test BézierFlow in the extreme low-NFE regime and identify where quality collapse begins, we conduct additional experiments in the very low-NFE range (NFE ≤ 3), which is even lower than the NFEs used in Sec. 5. Except for the NFEs, all other experiment setups follow those used in Sec. 5.

Table 9: **Quantitative comparison of few-step generation on text–image alignment with Stable Diffusion (Esser et al., 2024).** Results for the base solvers are reported on each top rows. **Bold** indicates the best results, and underline marks the second best. Gray cells indicate the base ODE solvers.

| Method | NFE=4 | | NFE=6 | | NFE=8 | | NFE=10 | |
|---|---|---|---|---|---|---|---|---|
| | CLIP ↑ | PickScore ↑ | CLIP ↑ | PickScore ↑ | CLIP ↑ | PickScore ↑ | CLIP ↑ | PickScore ↑ |
| MS-COCO 512 × 512 with Stable Diffusion (Esser et al., 2024) | | | | | | | | |
| RK1 | 0.240 | 0.206 | 0.252 | 0.212 | 0.257 | 0.215 | **0.260** | **0.217** |
| + DMN | 0.225 | 0.199 | 0.246 | 0.209 | 0.253 | 0.213 | 0.256 | 0.215 |
| + Bespoke | 0.241 | 0.206 | 0.243 | 0.212 | 0.251 | 0.214 | 0.252 | 0.216 |
| + GITS | 0.234 | 0.204 | 0.247 | 0.210 | 0.252 | 0.213 | 0.255 | 0.214 |
| + LD3 | 0.244 | 0.208 | 0.249 | 0.212 | **0.258** | **0.217** | 0.258 | **0.217** |
| + BézierFlow | **0.245** | **0.209** | **0.253** | **0.214** | 0.256 | **0.217** | 0.258 | **0.217** |
| RK2 | 0.244 | 0.208 | 0.255 | 0.214 | 0.259 | 0.216 | 0.260 | 0.217 |
| + DMN | 0.243 | 0.208 | 0.257 | **0.216** | 0.252 | 0.213 | 0.259 | 0.217 |
| + Bespoke | 0.244 | 0.208 | 0.225 | 0.200 | 0.253 | 0.215 | 0.257 | 0.217 |
| + GITS | **0.251** | **0.211** | 0.255 | 0.214 | 0.257 | 0.216 | 0.258 | 0.216 |
| + LD3 | 0.241 | 0.208 | 0.255 | 0.215 | **0.260** | **0.218** | 0.261 | 0.218 |
| + BézierFlow | 0.248 | 0.210 | **0.258** | 0.215 | **0.260** | 0.217 | **0.263** | **0.219** |

Table 10: **Quantitative comparison on training efficiency in few-step generation for diffusion and flow models on CIFAR-10.** All experiments are conducted on A6000 GPUs, except for the last row of distillation methods, which reports the performance of pretrained model from their official implementations (Song et al., 2023; Liu et al., 2023). "Time" denotes wall-clock training time, where s/m/d denote seconds/minutes/days, respectively.

| Method | NFE=6 | | NFE=8 | | Method | NFE=6 | | NFE=8 | |
|---|---|---|---|---|---|---|---|---|---|
| | FID ↓ | Time ↓ | FID ↓ | Time ↓ | | FID ↓ | Time ↓ | FID ↓ | Time ↓ |
| (1) Non-distillation Methods | | | | | | | | | |
| iPNDM | CIFAR-10 with EDM (Teacher FID: 2.08) | | | | RK2 | CIFAR-10 with ReFlow (Teacher FID: 2.70) | | | |
| + DMN | 9.33 | 5s | 4.82 | 5s | + DMN | 51.99 | 5s | 21.43 | 5s |
| + GITS | 6.80 | 30s | 4.07 | 30s | + GITS | 11.84 | 30s | 8.77 | 30s |
| + Bespoke | - | - | - | - | + Bespoke | 64.87 | 30m | 16.67 | 30m |
| + LD3 | 4.42 | 10m | 2.93 | 13m | + LD3 | 13.82 | 10m | 6.26 | 13m |
| + BézierFlow | 3.35 | 10m | 2.81 | 13m | + BézierFlow | 6.00 | 10m | 4.31 | 13m |
| (2) Distillation Methods | | | | | | | | | |
| CD | 359.59 | 15m | 343.59 | 15m | + 2-RF | 12.12 | 15m | 9.17 | 15m |
| CD | 4.24 | 6d | 3.95 | 6d | + 2-RF | 5.69 | 2d | 5.45 | 2d |
| CD | 2.82 | 8d (A100) | 2.79 | 8d (A100) | + 2-RF | 3.74 | 8d (A100) | 3.68 | 8d (A100) |

As summarized in Tab. 7 and Tab. 8, BézierFlow remains effective even at extremely low NFEs for both diffusion and flow models, improving over the base solvers by a substantial margin. Note that blank entries for RK2 simply reflect that RK2 only supports even numbers of function evaluations and has no timestep to learn in NFE=2. Furthermore, for NFE=1, timestep-learning methods cannot be applied, whereas scheduler-learning approaches such as Bespoke Solver (Shaul et al., 2024) and BézierFlow remain applicable.

## E.2 TEXT–IMAGE ALIGNMENT FOR FOUNDATIONAL MODEL

To complement the zero-shot MS-COCO FID results of Stable Diffusion v3.5 (Esser et al., 2024) in Tab. 2, we provide additional evaluation results for a more comprehensive assessment. We report CLIP score (Hessel et al., 2021) and PickScore (Kirstain et al., 2023), both of which measure the alignment between the given text prompt and the generated image.

As shown in Tab. 9, BézierFlow achieves the best or second-best performance across various NFEs, solvers, and evaluation metrics except for the CLIP Score at NFE=8 with the RK1 solver. These additional results further corroborate the superiority of BézierFlow even with the large-scale 2.5B pretrained stochastic interpolant model (Esser et al., 2024).

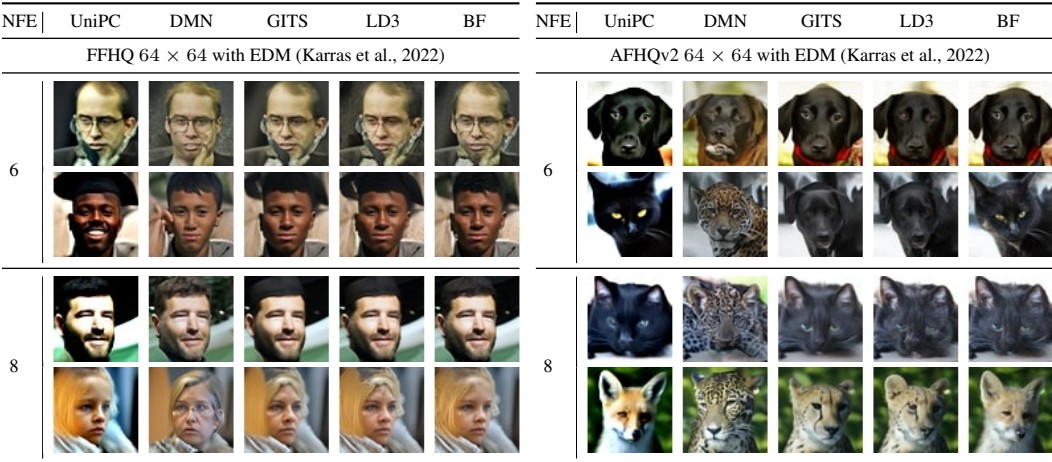

Figure 5: **Qualitative comparisons of samples generated using NFEs 6 and 8 on FFHQ and AFHQv2 datasets.** We use UniPC solver as the base solver for both cases.

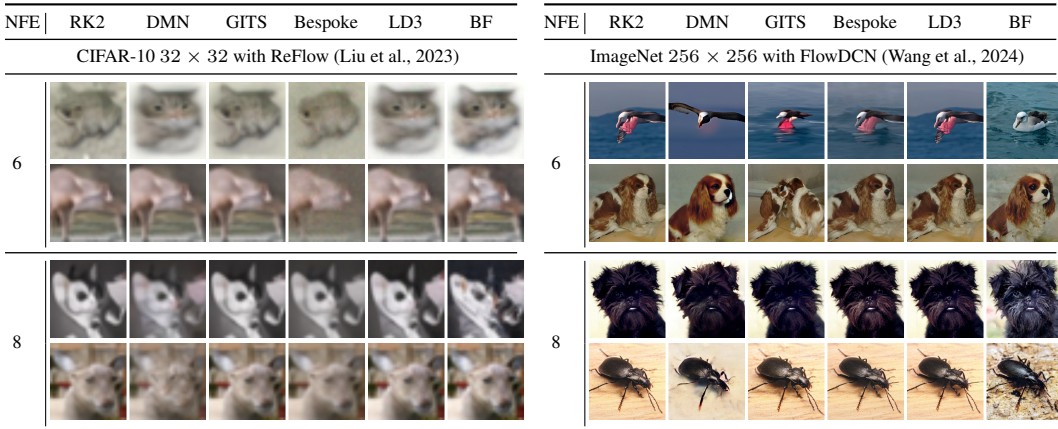

Figure 6: **Qualitative comparisons of samples generated using NFEs 6 and 8 on CIFAR-10 and ImageNet datasets.** We use RK2 solver as the base solver for both cases.

### E.3 COMPARISON ON TRAINING EFFICIENCY WITH FEW-STEP GENERATION METHODS

For a more comprehensive and fair comparison of training efficiency beyond the Tab. 4, we report additional results at matched NFEs with varying training budgets in Tab. 10. As shown, under the same NFEs, distillation-based approaches (Consistency Distillation (CD) (Song et al., 2023) and 2-Rectified Flow (2-RF) (Liu et al., 2023)) yield notably worse FID under the same lightweight training budget (15 minutes) and require *substantially longer* training time (2-6 days) to achieve FID comparable to BézierFlow, corresponding to roughly **200-600× more training time**. These results underscore BézierFlow's highly training-efficient acceleration, achieving in just a few minutes the performance that prior distillation-based approaches require several days of training to reach. Note that the 15-minute performance of 2-RF is identical to that of the base pretrained model as this budget is fully spent on the data creation stage for ReFlow.

We also include training time comparisons against non-distillation baselines that accelerate generation with lightweight training, including DMN, GITS, Bespoke Solver and LD3 (Xue et al., 2024; Chen et al., 2024; Shaul et al., 2024; Tong et al., 2025). Among these lightweight acceleration methods, BézierFlow achieves the best FID, even outperforming LD3 under the same training budget. This demonstrates that BézierFlow offers a more favorable trade-off between training efficiency and sample quality.

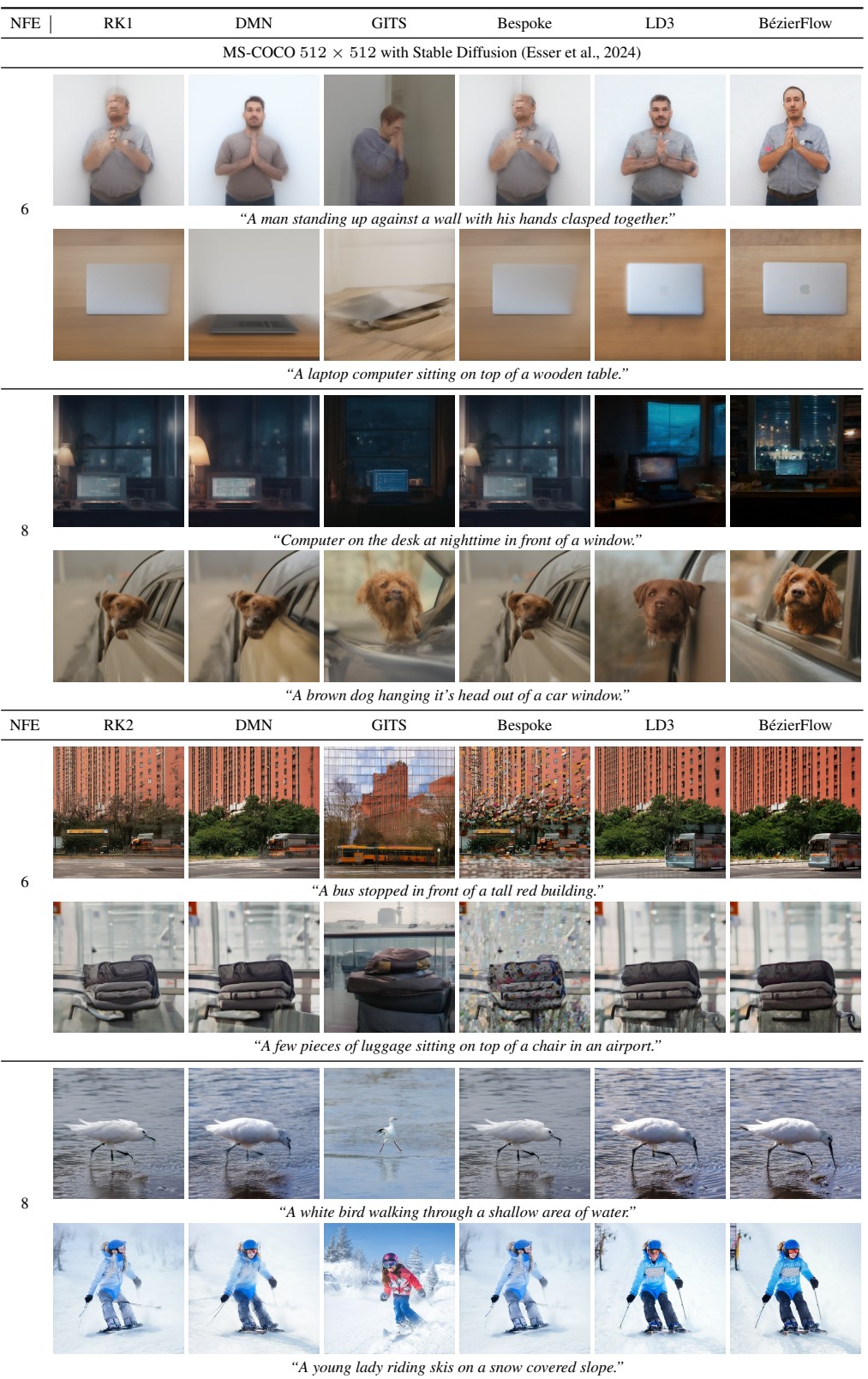

Figure 7: **Qualitative comparisons of samples generated using NFEs 6 and 8 with Stable Diffusion v3.5 Esser et al. (2024).** We use RK1 and RK2 as the base solver.

Table 11: **Quantitative comparison on unconditional 3D point cloud generation with Point Voxel Diffusion (PVD) (Zhou et al., 2021).** Lower is better for CD-MMD (denoted as MMD) and JSD and higher is better for CD-COV (denoted as COV). CD-MMD is multiplied by $10^3$. Results for the base solvers are reported on each top rows. **Bold** indicates the best results, and underline marks the second best. Gray cells indicate the base ODE solvers.

| Method | NFE=4 | | | NFE=6 | | | NFE=8 | | | NFE=10 | | |
|---|---|---|---|---|---|---|---|---|---|---|---|---|
| | MMD ↓ | COV ↑ | JSD ↓ | MMD ↓ | COV ↑ | JSD ↓ | MMD ↓ | COV ↑ | JSD ↓ | MMD ↓ | COV ↑ | JSD ↓ |
| UniPC | 2.50 | 3.21 | 0.46 | 1.25 | 8.89 | 0.30 | 0.95 | 17.03 | 0.25 | 0.79 | 20.25 | 0.22 |
| + DMN | 1.10 | 16.79 | 0.27 | 0.68 | **26.42** | **0.23** | 1.50 | 15.06 | 0.32 | 0.67 | 19.51 | 0.27 |
| + GITS | 5.32 | 9.52 | 0.56 | 9.14 | 0.74 | 0.63 | 1.20 | 20.25 | 0.31 | 0.90 | 16.30 | 0.23 |
| + LD3 | 1.20 | **21.23** | **0.24** | 1.16 | 20.74 | 0.24 | 0.80 | 21.48 | 0.25 | 0.91 | 21.23 | 0.23 |
| + BézierFlow | **0.88** | 18.77 | 0.29 | **0.59** | 22.72 | **0.23** | 0.58 | **23.45** | **0.24** | **0.53** | **23.70** | **0.21** |
| iPNDM | 1.17 | 14.81 | **0.27** | 0.91 | 16.54 | 0.23 | 0.78 | 23.46 | **0.21** | 0.67 | **26.67** | **0.20** |
| + DMN | 1.18 | **19.26** | 0.29 | 0.63 | **24.69** | 0.22 | 1.74 | 6.17 | 0.35 | 0.65 | 20.49 | 0.22 |
| + GITS | 3.22 | 7.65 | 0.44 | 3.59 | 3.21 | 0.48 | 3.99 | 1.73 | 0.49 | 2.73 | 3.95 | 0.41 |
| + LD3 | 2.40 | 13.33 | 0.34 | 0.89 | 18.52 | 0.25 | 0.77 | 19.01 | 0.25 | 0.70 | 22.72 | 0.22 |
| + BézierFlow | **0.85** | 18.52 | 0.29 | **0.58** | 22.73 | 0.23 | **0.57** | **24.44** | 0.23 | **0.56** | 24.52 | 0.21 |

## F  MORE QUALITATIVE RESULTS

We provide more qualitative results for accelerated sampling of diffusion models in Fig. 5 and flow models in Fig. 6 and Fig. 7. Across both model classes, BézierFlow (BF) consistently yields clearer structures and more faithful details compared to baselines under low NFEs.

## G  EXTENSION TO OTHER DOMAINS

BézierFlow is a generic framework applicable not only to image synthesis but also to various generative tasks within the stochastic interpolant framework. To demonstrate the versatility of our method and its robustness to different distance metrics beyond LPIPS, we conduct additional experiments on two distinct domains: 3D point cloud generation and layout generation.

### G.1  UNCONDITIONAL 3D POINT CLOUD GENERATION

3D Point cloud generation involves creating 3D representations of objects using discrete points, a task essential for applications in robotics, autonomous driving, and 3D modeling. We evaluate BézierFlow using the Point-Voxel Diffusion (PVD) model (Zhou et al., 2021), trained on the *airplane* category of the ShapeNet dataset (Chang et al., 2015).

**Experiment Setup.** We adopt a simple mean squared error (MSE) loss for both training and validation. We generate 32 noise–data pairs for both the training and validation sets using DPM-Solver (Lu et al., 2022) with 64 NFEs, and train the model for 5 epochs. We compare our method against the same set of baselines reported in Tab. 1.

**Evaluation Metrics.** Following the evaluation protocol of PVD (Zhou et al., 2021), we assess the quality of generated samples using three metrics based on the Chamfer Distance (CD): Minimum Matching Distance (CD-MMD), Coverage Score (CD-COV), and Jensen-Shannon Divergence (JSD).

**Results.** Tab. 11 presents the quantitative results. BézierFlow consistently achieves the best or second-best performance on CD-MMD, CD-COV across all NFEs, substantially improving over the base solvers and timestep-learning baselines (Xue et al., 2024; Chen et al., 2024; Tong et al., 2025). Fig. 8 provides qualitative comparisons of generated 3D point clouds, where BézierFlow better preserves both the global shape and coverage of the target distribution.

### G.2  UNCONDITIONAL LAYOUT GENERATION

Layout generation aims to synthesize structural arrangements of elements (e.g., UI components, document blocks), which is a critical step in graphic design automation. We evaluate our method on

| NFE | iPNDM | DMN | GITS | LD3 | BézierFlow |
|---|---|---|---|---|---|
| | ShapeNet *airplane* with PVD (Zhou et al., 2021) | | | | |

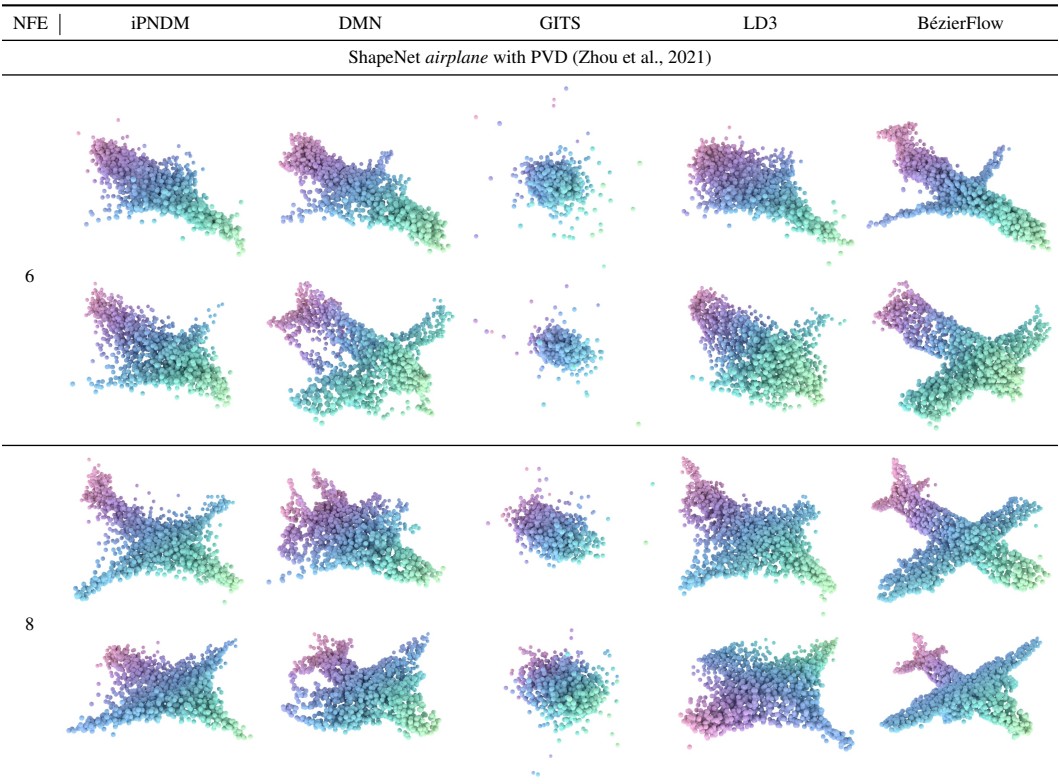

Figure 8: **Qualitative comparisons of 3D point cloud samples generated using NFEs 6 and 8 with PVD (Zhou et al., 2021).** We use iPNDM as the base solver.

Table 12: **Quantitative comparison on unconditional layout generation with Layout-Flow (Guerreiro et al., 2024).** Lower is better for FID, Alignment (denoted as Align.), Overlap. Results for the base solvers are reported on each top rows. **Bold** indicates the best results, and underline marks the second best. Gray cells indicate the base ODE solvers.

| Method | NFE=4 | | | NFE=6 | | | NFE=8 | | | NFE=10 | | |
|---|---|---|---|---|---|---|---|---|---|---|---|---|
| | FID ↓ | Align. ↓ | Overlap ↓ | FID ↓ | Align. ↓ | Overlap ↓ | FID ↓ | Align. ↓ | Overlap ↓ | FID ↓ | Align. ↓ | Overlap ↓ |
| RK1 | 55.88 | 0.40 | 0.60 | 22.75 | 0.35 | 0.56 | 11.66 | 0.30 | 0.54 | 7.93 | 0.27 | 0.52 |
| + DMN | 178.35 | 0.55 | 1.08 | 88.40 | 0.69 | 0.70 | 26.27 | 0.37 | 0.46 | 10.96 | 0.33 | **0.46** |
| + GITS | 41.08 | 0.37 | 0.57 | 12.84 | 0.35 | **0.47** | 7.32 | 0.29 | **0.45** | 5.90 | 0.27 | **0.46** |
| + Bespoke | 213.61 | 0.92 | 1.01 | 201.20 | 0.88 | 0.67 | 168.49 | 0.63 | 0.59 | 171.11 | 0.63 | 0.56 |
| + LD3 | **19.51** | **0.32** | 0.54 | 8.36 | 0.28 | 0.51 | 5.03 | **0.23** | 0.48 | 3.70 | 0.23 | 0.47 |
| + BézierFlow | 32.78 | 0.35 | **0.53** | **7.10** | **0.26** | 0.47 | **3.86** | 0.25 | 0.49 | **2.96** | **0.22** | 0.50 |
| RK2 | 143.90 | 0.67 | 0.65 | 73.91 | 0.47 | 0.49 | 35.84 | 0.38 | 0.51 | 20.80 | 0.34 | 0.51 |
| + DMN | 142.40 | 0.66 | **0.35** | 88.15 | 0.49 | 0.46 | 63.57 | 0.37 | **0.42** | 56.23 | 0.35 | **0.43** |
| + GITS | **102.11** | **0.49** | 0.42 | 51.62 | 0.37 | 0.48 | 27.84 | 0.32 | 0.50 | 8.25 | **0.22** | 0.47 |
| + Bespoke | 126.80 | 0.61 | 0.47 | 187.62 | 0.86 | **0.37** | 32.99 | 0.38 | 0.54 | 21.54 | 0.36 | 0.50 |
| + LD3 | 162.98 | 0.62 | 0.47 | 42.82 | 0.37 | 0.48 | **12.57** | **0.26** | 0.48 | 8.39 | 0.27 | 0.48 |
| + BézierFlow | 142.34 | 0.63 | 0.57 | **39.17** | **0.35** | 0.52 | 25.51 | 0.37 | 0.50 | **7.18** | 0.26 | 0.49 |

unconditional layout generation using LayoutFlow (Guerreiro et al., 2024), pretrained on the RICO dataset (Deka et al., 2017).

**Experiment Setup.** We adopt negative mean Intersection over Union (mIoU) between the teacher and student layouts as the objective for both training and validation. We generate 50 noise–data pairs for both the training and validation sets using an RK45 (Butcher, 1996) solver, and train the model for 5 epochs. We compare our method against the same set of baselines reported in Tab. 2.

**Evaluation Metrics.** Following the evaluation protocol of LayoutFlow (Guerreiro et al., 2024), we assess generation quality using Fréchet Inception Distance (FID) adapted for layouts, alongside

| NFE | RK1 | DMN | GITS | Bespoke | LD3 | BézierFlow | Teacher |
|---|---|---|---|---|---|---|---|

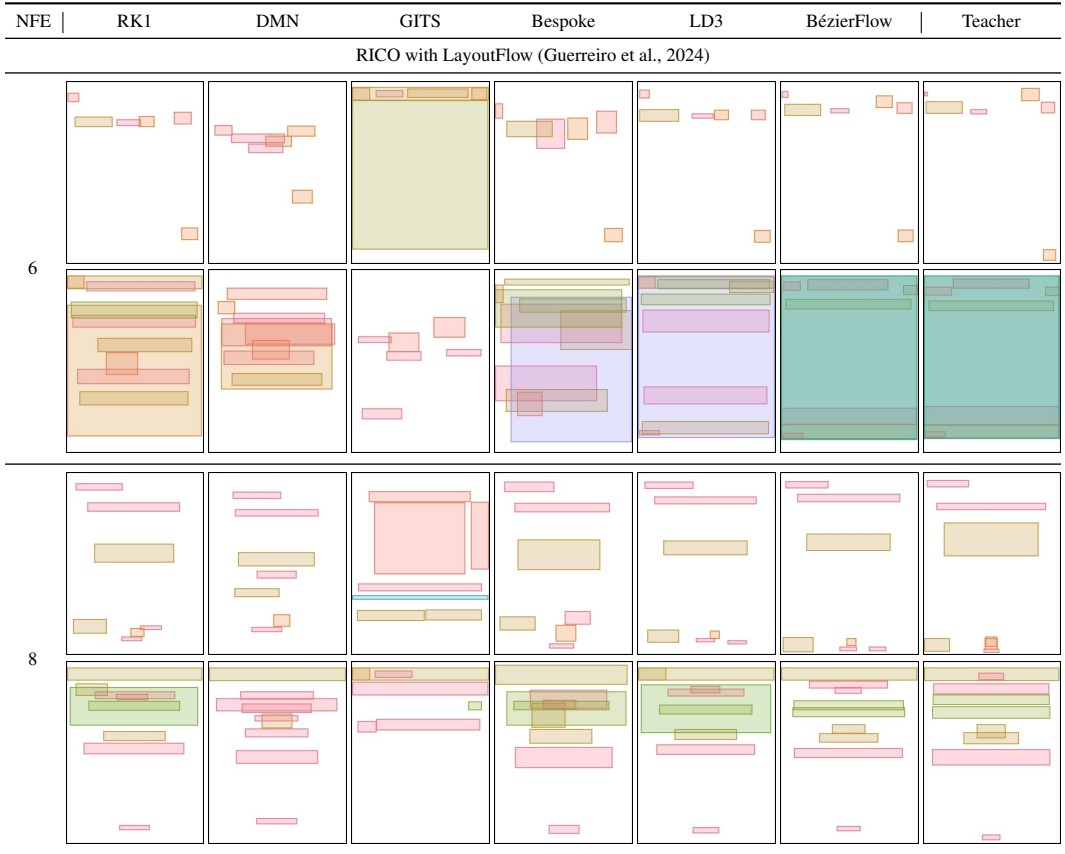

Figure 9: **Qualitative comparisons of layout samples generated using NFEs 6 and 8 with LayoutFlow (Guerreiro et al., 2024).** We use RK1 as the base solver. The rightmost column shows teacher samples from RK45 solver.

Table 13: **FID comparison of VDM, Multi-marginal SI (denoted as MMSI), and BézierFlow on CIFAR-10.** Results for the base solvers are reported on each top rows. **Bold** indicates the best results, and underline marks the second best. Gray cells indicate the base ODE solvers.

| Method | NFE=4 | NFE=6 | NFE=8 | NFE=10 | Method | NFE=4 | NFE=6 | NFE=8 | NFE=10 |
|---|---|---|---|---|---|---|---|---|---|
| CIFAR-10 32 $\times$ 32 with ReFlow (Liu et al., 2023) (Teacher FID: 2.70) | | | | | | | | | |
| RK1 | 52.78 | 26.30 | 17.40 | 13.30 | RK2 | 25.36 | 12.12 | 9.17 | 7.89 |
| + VDM | 54.72 | 22.06 | 19.10 | 19.00 | + VDM | 36.24 | 25.74 | 16.37 | 12.39 |
| + MMSI | 22.89 | 12.06 | 7.59 | 5.86 | + MMSI | 20.82 | 9.03 | 7.57 | 7.79 |
| + BézierFlow | **20.64** | **9.67** | **7.30** | **5.51** | + BézierFlow | **13.18** | **6.00** | **4.31** | **3.74** |

Alignment and Overlap scores. For FID calculation, we employ the feature extractor from Layout-Diffusion (Zheng et al., 2023).

**Results.** As shown in Table 12, BézierFlow shows better performance than other baselines in terms of FID and Alignment. Interestingly, it consistently outperforms the base solvers (RK1 and RK2) on all metrics at the same NFE settings. We provide a qualitative comparison of the generated layouts in Fig. 9. BézierFlow produces layouts that most closely follow the teacher trajectory, preserving the spatial arrangement and aspect ratios of objects.

## H  COMPARISON WITH OTHER SCHEDULER PARAMETERIZATIONS

In this section, we discuss prior work (Kingma et al., 2023; Albergo et al., 2024) that also learns optimal SI schedulers and compare them against BézierFlow. We first clarify how these methods differ in their scheduler parameterizations, and then experimentally show that BézierFlow achieves superior performance due to its compact parameterization that explicitly satisfies the core SI scheduler requirements: boundary conditions, monotonicity, and differentiability.

**Varitional Diffusion Models (Kingma et al., 2023).**  Variational Diffusion Models (VDMs) model the signal-to-noise ratio function $\mathrm{SNR}(t)$ with a monotone neural network to satisfy monotonicity, aiming primarily to improve generative performance rather than sampling acceleration. However, this neural network contains more than 1024 parameters, and thus is not parameter-efficient. In contrast, BézierFlow uses a much more compact parameterization with only $n = 32$ control points in our experiments by leveraging 1-D Bézier functions, reducing the number of scheduler parameters by roughly an order of magnitude.

**Multi-Marginal Stochastic Interpolant (Albergo et al., 2024).**  Multi-Marginal Stochastic Interpolant (Multi-Marginal SI) also learns a stochastic interpolant scheduler to improve generative performance. In the 2-marginal case, the (unnormalized) scheduler is parameterized as

$$\tilde{\alpha}(s) = 1 - s + \left(\sum_{k=1}^{K} a_k \sin\left(\frac{\pi}{2}t\right)\right)^2, \qquad \tilde{\sigma}(s) = s + \left(\sum_{k=1}^{K} b_k \sin\left(\frac{\pi}{2}t\right)\right)^2, \qquad (39)$$

with learnable coefficients $a_k$ and $b_k$, which are then normalized via

$$\bar{\alpha}(s) = \frac{\tilde{\alpha}(s)}{\tilde{\alpha}(s) + \tilde{\sigma}(s)}, \qquad \bar{\sigma}(s) = \frac{\tilde{\sigma}(s)}{\tilde{\alpha}(s) + \tilde{\sigma}(s)}. \qquad (40)$$

While this parameterization enforces the boundary conditions $\bar{\alpha}(0) = 0$, $\bar{\sigma}(1) = 0$, $\bar{\alpha}(1) = 1$, and $\bar{\sigma}(0) = 1$, the induced SNR schedule $\bar{\rho}(s) = \bar{\alpha}(s)/\bar{\sigma}(s)$ is not guaranteed to be monotonically increasing. In contrast, our Bézier-based parameterization explicitly satisfies the three core requirements of an SI scheduler: (1) boundary conditions, (2) monotonicity, and (3) differentiability. This advantage is reflected in the quantitative comparison reported below.

**Results.**  We report few-step generation FIDs on CIFAR-10 with ReFlow (Liu et al., 2023) for VDM (Kingma et al., 2023), Multi-Marginal SI (Albergo et al., 2024), and BézierFlow. For VDM, we parameterize the SNR neural network as a 3-layer MLP with hidden size 1024, following the original configuration in the paper, and set the trigonometric order of Multi-Marginal SI to $K = 32$ so that its number of scheduler degrees of freedom matches our $n = 32$ Bézier parameterization. As shown in Tab. 13, BézierFlow consistently achieves the best FID across all NFEs and base solvers, outperforming VDM and Multi-Marginal SI under the same training setup. This highlights that our Bézier-based parameterization, which satisfies the key requirements of an SI scheduler, provides a more effective and stable way to learn SI schedulers for few-step generation than existing neural or trigonometric alternatives.

## I  CROSS-DATASET TRANSFER OF BÉZIER SCHEDULER

We investigate whether a BézierFlow trained on one dataset can be reused on other datasets without retraining. Specifically, we train BézierFlow on CIFAR-10 with pretrained diffusion models (Karras et al., 2022) and then evaluate on two different datasets, FFHQ and AFHQv2. In Tab. 14, we report FIDs for the base ODE solvers, the dataset-specific scheduler (denoted as "BézierFlow") and the CIFAR-10–trained scheduler reused on the target datasets (denoted as "Transferred").

As shown, although the scheduler is trained only on CIFAR-10, its performance on out-of-domain datasets still outperforms the base solvers and remains competitive with a scheduler trained directly on the target dataset. This demonstrates that BézierFlow provides a generally effective acceleration scheme even under domain shift.

Table 14: **Cross-dataset transfer of Bézier stochastic interpolant schedulers.** Results for the base solvers are reported on each top rows. **Bold** indicates the best results, and underline marks the second best. Gray cells indicate the base ODE solvers.

| Method | NFE=4 | NFE=6 | NFE=8 | NFE=10 | Method | NFE=4 | NFE=6 | NFE=8 | NFE=10 |
|---|---|---|---|---|---|---|---|---|---|
| CIFAR-10 32 × 32 with EDM → FFHQ 64 × 64 with EDM | | | | | | | | | |
| UniPC | 47.62 | 14.96 | 7.76 | 8.93 | iPNDM | 28.75 | 11.15 | 6.68 | 4.80 |
| + BézierFlow | **17.05** | **7.43** | **3.82** | **3.13** | + BézierFlow | **15.39** | **7.84** | 5.56 | **3.75** |
| + Transferred | 22.35 | 9.05 | 4.93 | 4.50 | + Transferred | 23.41 | 9.87 | **5.47** | 3.79 |
| CIFAR-10 32 × 32 with EDM → AFHQv2 64 × 64 with EDM | | | | | | | | | |
| UniPC | 23.59 | 10.15 | 7.76 | 6.38 | iPNDM | 15.14 | 6.12 | 3.80 | 3.01 |
| + BézierFlow | 12.27 | **4.46** | **2.75** | **2.67** | + BézierFlow | 14.44 | 4.69 | **2.63** | **2.16** |
| + Transferred | **11.58** | 4.47 | 2.98 | 2.71 | + Transferred | **9.52** | **3.95** | 2.96 | 2.30 |

Table 15: **Training time and peak GPU memory usage of BézierFlow for diffusion and flow models at NFEs 4 and 10 on a single A6000 GPU.**

| Dataset / Model | NFE=4 | | NFE=10 | |
|---|---|---|---|---|
| | Training Time | VRAM | Training Time | VRAM |
| (1) Diffusion Models | | | | |
| CIFAR-10 32×32 with EDM (Karras et al., 2022) | 8 minutes | 4 GB | 15 minutes | 8 GB |
| FFHQ 64×64 with EDM (Karras et al., 2022) | 11 minutes | 3 GB | 18 minutes | 7 GB |
| AFHQv2 64×64 with EDM (Karras et al., 2022) | 11 minutes | 3 GB | 18 minutes | 7 GB |
| (2) Flow Models | | | | |
| CIFAR-10 32×32 with ReFlow (Liu et al., 2023) | 8 minutes | 3 GB | 15 minutes | 7 GB |
| ImageNet 256×256 with FlowDCN (Wang et al., 2024) | 25 minutes | 5 GB | 45 minutes | 8 GB |
| MS-COCO 512×512 with Stable Diffusion (Esser et al., 2024) | 60 minutes | 21 GB | 100 minutes | 22 GB |

## J   COMPUTATIONAL COSTS

We report wall-clock training time and peak GPU memory for BézierFlow across all datasets and both diffusion and flow models, evaluated at NFE=4 and NFE=10 on a single A6000 GPU. As shown in Tab. 15, BézierFlow trains in at most 1 hour even for a 2.5B large-scale pretrained model (Esser et al., 2024) at $512 \times 512$ resolution, while requiring only 22 GB of GPU memory. This makes the method practical even on a single commodity GPU commonly available in research labs. Despite this low training and memory cost, BézierFlow improves FID by large margins over the base model, e.g., **from 50.30 to 9.55 (≈81% relative improvement)** at NFE=4 on CIFAR-10 and **from 8.93 to 3.13 (≈64% relative improvement)** at NFE=10 on FFHQ, as shown in Tab. 1.

The increase in training time from NFE=4 to NFE=10 is always less than $2\times$, and peak GPU memory grows only mildly with NFE. This is because we apply gradient checkpointing over the student trajectory, so activation memory scales only weakly with the number of steps, even for Stable Diffusion v3.5. These results indicate that BézierFlow scales to high-resolution, large models with modest computational overhead, making it practical as a plug-and-play scheduler even for large-scale generative models.

