# OpenReview forum: "BézierFlow: Learning Bézier Stochastic Interpolant Schedulers for Few-Step Generation"
_ICLR.cc/2026/Conference — ICLR 2026 Poster_

### Official Review · Reviewer_2baL · 2025-10-31

**Soundness:** 3
**Presentation:** 4
**Contribution:** 3
**Rating:** 8
**Confidence:** 3

**Summary:**

The paper proposes BézierFlow (BF), a lightweight, training-efficient way to learn a scheduler for Stochastic-Interpolant (SI) models so that few-step ODE sampling matches a strong many-step teacher. Schedulers $\left(\bar{\alpha},\bar{\sigma}\right)$ are parameterized as low-dimensional Bézier curves and optimized via learnable control points against the teacher trajectory. BézierFlow demonstrate high-quality few-step sampling ($\approx$3–8 NFEs) with minimal tuning and no model retraining.

**Strengths:**

– Writting is clear and easy to follow.

– Using Bézier control points to model SI schedulers is simple and effective.

– Competitive results vs. baselines at very low NFE.

**Weaknesses:**

– Although the target problem differs, VDM [1] and Multi-marginal SI [2] also optimize schedules to improve performance. A brief discussion contrasting these with BézierFlow would clarify the advantages of using Bézier curves versus alternative parameterizations (and the trade-offs without Bézier).

– I’m curious about the method’s limits. Recent distillation works [3,4] reduce sampling to a few or even one step while maintaining quality. I understand BF is lightweight and not the same setting, but probing the few-to-one NFE regime would better reveal the method’s capability.

[1] Kingma et al., “Variational Diffusion Models”, NeurIPS 2021

[2] Albergo et al., “Multimarginal generative modeling with stochastic interpolants”, NeurIPS 2024

[3] Zhou et al., “Inductive moment matching”, ICML 2025

[4] Kim et al., “Consistency trajectory models: Learning probability flow ode trajectory of diffusion”, ICLR 2024

**Questions:**

– How far can BF push NFE down (e.g., $\le2$) before quality collapses?

– Does a scheduler learned on dataset A transfer to B (or to different guidance scales/resolutions) without re-tuning?

---

> ### Author Response · Authors · 2025-11-25
> **Response to Reviewer 2baL (1/2)**
>
> Dear **Reviewer 2baL**,
>
> We appreciate the reviewer’s positive assessment and are glad that you found **our Bézier parameterization simple and effective**. We also thank you for the constructive and insightful feedback regarding its connection to alternative parameterizations, the behavior of the method in the extreme few-to-one NFE regime, and the transferability of BézierFlow across different datasets. Below, we address each of the points you raised.
>
> ---
> ### W1. Relation with VDM and Multi-Marginal SI
>
> Thanks for bringing to our attention additional related work. Within the lens of the Stochastic Interpolant (SI) framework, all of them identically optimize the scheduler $(\bar\alpha(s), \bar\sigma(s))$, but differ in how it is parameterized. VDM, for instance, models $\text{SNR}(t)$ with a monotone neural network using more than 1024 parameters, whereas BézierFlow leverages 1-D Bézier functions that represent the SNR trajectory with a much more compact set of parameters (32 control points used in the main paper). Beyond this compactness, the strength of our parameterization is further highlighted when compared to Multi-Marginal SI: although Multi-Marginal SI also optimizes the scheduler, its parameterization does not enforce monotonicity. In contrast, our Bézier-based parameterization explicitly satisfies the three core requirements of an SI scheduler: (1) boundary conditions, (2) monotonicity, and (3) differentiability. This advantage is reflected in the quantitative comparison in the table below, where BézierFlow outperforms these approaches across different NFEs. We include this discussion and the results in App. H of the revised manuscript.
>
>
> \begin{array}{lccccc}
> \hline
> \text{Method}&\text{NFE=4}&\text{NFE=6}&\text{NFE=8}&\text{NFE=10}&
> \newline
> \hline
> \text{RK1}
> \newline
> \text{+ VDM}&54.72&22.06&19.10&19.00
> \newline
> \text{+ MMSI}&\underline{22.89}&\underline{12.06}&\underline{7.59}&\underline{5.86}
> \newline
> \text{+ BézierFlow}&\textbf{20.64}&\textbf{9.67}&\textbf{7.30}&\textbf{5.51}
> \newline
> \hline
> \text{RK2}
> \newline
> \text{+ VDM}&36.24&25.74&16.37&12.39
> \newline
> \text{+ MMSI}&\underline{20.82}&\underline{9.03}&\underline{7.57}&\underline{7.79}
> \newline
> \text{+ BézierFlow}&\textbf{13.18}&\textbf{6.00}&\textbf{4.31}&\textbf{3.74}
> \newline
> \hline
> \end{array}
>
> ---
> ### W2,Q1. Probing BézierFlow at Extremely Low NFEs
>
> Following the reviewer’s suggestion, we also conduct experiments at even lower NFEs (1-3) than those considered in the main paper (NFE=4). The results show that BézierFlow remains effective in this extreme regime, improving over the base solver even at NFE$\le 3$ with only a few minutes of extra training. We include more comprehensive results for the extremely low-NFE setting in App. E.1 of the revised manuscript.
>
>
> \begin{array}{lcccc}
> \hline
> \text{Method}&\text{NFE}=1&\text{NFE}=2&\text{NFE}=3&\text{NFE}=1&\text{NFE}=2&\text{NFE}=3
> \newline
> \text{}&\text{CIFAR-10 FID}\downarrow&\text{CIFAR-10 FID}\downarrow&\text{CIFAR-10 FID}\downarrow&\text{FFHQ FID}\downarrow&\text{FFHQ FID}\downarrow&\text{FFHQ FID}\downarrow\newline
> \hline
> \text{UniPC}&\underline{377.15}&168.35&57.45&\underline{280.61}&104.57&59.54\newline
> \text{+ DMN}&-&\underline{160.65}&66.03&-&142.57&64.99\newline
> \text{+ GITS}&-&168.29&\underline{53.21}&-&\underline{107.71}&\underline{42.38}\newline
> \text{+ LD3}&-&187.42&\textbf{39.56}&-&120.87&48.30\newline
> \textbf{+ BézierFlow (Ours)}&\textbf{125.03}&\mathbf{50.41}&55.07&\textbf{121.94}&\mathbf{72.03}&\mathbf{33.72}\newline
> \hline
> \text{iPNDM}&\underline{377.15}&153.31&47.68&\underline{280.61}&102.50&45.70\newline
> \text{+ DMN}&-&146.40&58.98&-&112.55&61.54\newline
> \text{+ GITS}&-&153.33&43.71&-&105.78&\textbf{32.33}\newline
> \text{+ LD3}&-&\underline{145.03}&\underline{32.19}&-&\underline{97.62}&38.14\newline
> \textbf{+ BézierFlow (Ours)}&\textbf{125.03}&\mathbf{41.58}&\mathbf{22.20}&\textbf{121.94}&\mathbf{60.45}&\underline{35.10}\newline
> \hline
> \end{array}

---

> > ### Author Response · Authors · 2025-11-25
> > **Response to Reviewer 2baL (2/2)**
> >
> > ### Q2. Cross-Dataset Transfer of Bézier Stochastic Interpolant Scheduler
> > Thanks for the insightful suggestion. We evaluate cross-dataset transfer by applying a scheduler trained on CIFAR-10 to different datasets. Even though the model is trained only on CIFAR-10, its performance on out-of-domain datasets still outperforms the base model and, while slightly below that of a scheduler trained on the target dataset, remains competitive. This suggests that BézierFlow provides a generally effective acceleration scheme even under domain shift. We add these cross-dataset transfer results in App. I of the revised manuscript.
> >
> >
> > \begin{array}{lcccc}
> > \hline
> > \text{} & \text{NFE=4} & \text{NFE=6} & \text{NFE=8} & \text{NFE=10} \newline
> > \hline
> > \text{CIFAR-10 $32\times 32$ $\to$ FFHQ $64\times 64$} & & & & \newline
> > \hline
> > \text{UniPC}&47.62&14.96&7.76&8.93 \newline
> > \text{+ BézierFlow}        & \textbf{17.05} & \textbf{7.43} & \textbf{3.82} & \textbf{3.13} \newline
> > \text{+ Transferred} & \underline{22.35} & \underline{9.05} & \underline{4.93} & \underline{4.50} \newline
> > \hline
> > \text{iPNDM}&28.75&11.15&6.68&4.80 \newline
> > \text{+ BézierFlow}        & \textbf{15.39} & \textbf{7.84} & \underline{5.56} & \textbf{3.75} \newline
> > \text{+ Transferred} & \underline{23.41} & \underline{9.87} & \textbf{5.47} & \underline{3.79} \newline
> > \hline
> > \text{CIFAR-10 $32\times 32$ $\to$ AFHQv2 $64\times 64$} & & & & \newline
> > \hline
> > \text{UniPC}&23.59&10.15&7.76&6.38 \newline
> > \text{+ BézierFlow}        & \underline{12.27} & \textbf{4.46} & \textbf{2.75} & \textbf{2.67} \newline
> > \text{+ Transferred} & \textbf{11.58} & \underline{4.47} & \underline{2.98} & \underline{2.71} \newline
> > \hline
> > \text{iPNDM}&15.14&6.12&3.80&3.01\newline
> > \text{+ BézierFlow}        & \underline{14.44} & \underline{4.69} & \textbf{2.63} & \textbf{2.16} \newline
> > \text{+ Transferred} & \textbf{9.52}  & \textbf{3.95} & \underline{2.96} & \underline{2.30} \newline
> > \hline
> > \end{array}

---

> > > ### Comment · Reviewer_2baL · 2025-11-28
> > >
> > > Thank you for your detailed responses. My concerns have been basically addressed. I will keep my original score and support the acceptance of this paper.

---

> > > > ### Author Response · Authors · 2025-11-28
> > > > **Response to Reviewer 2baL**
> > > >
> > > > Dear Reviewer 2baL,
> > > >
> > > > We are happy to hear that our rebuttal addressed your concerns well. Also, we deeply appreciate your support for our work.
> > > >
> > > > Best,
> > > >
> > > > Authors

---

### Official Review · Reviewer_TYFB · 2025-10-31

**Soundness:** 3
**Presentation:** 3
**Contribution:** 3
**Rating:** 4
**Confidence:** 4

**Summary:**

The paper proposes BézierFlow, a lightweight method to improve few-step generation for pretrained diffusion and flow models by learning the sampling trajectory—formulated as a stochastic interpolant (SI) scheduler—instead of only optimizing discrete ODE timesteps. The scheduler’s coefficient functions are parameterized as 1D Bézier curves whose control points enforce boundary conditions, differentiability, and (claimed) monotonic SNR. Training is a teacher-forcing alignment to a high-NFE teacher using a perceptual loss, and takes ~minutes.

**Strengths:**

1. This paper moves beyond learned timesteps to continuous path learning via SI schedulers, which conceptually unifies diffusion and flow settings and widens the search space versus LD3.  Besides, it also provides analyses for endpoint-marginal preservation and schedule-invariance of the SI training objective.

2. This method only needs a few parameters and minutes of training, no finetuning of the base model, and it supports plug-and-play at inference.

**Weaknesses:**

1. Could the authors include more qualitative comparisons and broader evaluation metrics on SD3 models? Besides, for large-scale t2i model like SD3, except the FID values, other metrics, like CLIP are also important to evaluate the performance.

2. Notice that training relies solely on LPIPS. Could this induce instability or mode collapse by using only this loss across most models training?

3. The authors analyze briefly in Sec 4.4 the difference between BézierFlow and other schedulers, but this anlysis is not convincing to clarify its advantages over other schedulers. Could the authors provide more solid theoretical analysis?

4. While “15 minutes” training time is appealing, results are only on CIFAR-10. Include runtime and memory analysis on larger datasets and models (e.g., ImageNet, SD3) to support the scalability claim.

**Questions:**

see Weakness

---

> ### Author Response · Authors · 2025-11-25
> **Response to Reviewer TYFB (1/2)**
>
> Dear **Reviewer TYFB**,
>
> We thank the reviewer for the constructive and detailed feedback. Below we respond to your comments on Stable Diffusion evaluation metrics, LPIPS-based training stability, our comparison to other schedulers, and the scalability of BézierFlow.
>
> ---
> ### W1. More Evaluation Metrics for Stable Diffusion v3.5”
>
> Thanks for the suggestion to include additional evaluation metrics for SD3. For a more comprehensive assessment, we now report CLIP Score [A] and PickScore [B], both of which measure the alignment between the given text prompt and the generated image.
> The table below reports these metrics alongside the original zero-shot MS-COCO FID score reported in the main paper. As shown, BézierFlow achieves the best or second-best performance across various NFEs, solvers, and evaluation metrics except for the CLIP Score at NFE=8 with the RK1 solver. These additional results further corroborate the superiority of BézierFlow even with the large-scale pretrained diffusion model. Please refer to Fig. 7 in App. F of the revised manuscript for qualitative comparisons, where BézierFlow produces much sharper outputs than the baselines.
>
> \begin{array}{lccccccccccccc}
> \hline
> \text{Method}
>   &&\text{NFE=4}&&&\text{NFE=6}&&&\text{NFE=8}&&&\text{NFE=10}&&
> \newline
> \text{}
>   &\text{FID}\downarrow&\text{CLIP}\uparrow&\text{PickScore}\uparrow
>   &\text{FID}\downarrow&\text{CLIP}\uparrow&\text{PickScore}\uparrow
>   &\text{FID}\downarrow&\text{CLIP}\uparrow&\text{PickScore}\uparrow
>   &\text{FID}\downarrow&\text{CLIP}\uparrow&\text{PickScore}\uparrow
> \newline
> \hline
> \text{RK1}&57.93&0.240&0.206&\textbf{30.96}&\underline{0.252}&\underline{0.212}&21.50&\underline{0.257}&\underline{0.215}&17.19&\textbf{0.260}&\textbf{0.217}
> \newline
> \text{+ DMN}&113.24&0.225&0.199&46.02&0.246&0.209&31.58&0.253&0.213&24.41&0.256&0.215
> \newline
> \text{+ GITS}&70.01&0.234&0.204&42.44&0.247&0.210&31.89&0.252&0.213&25.47&0.255&0.214
> \newline
> \text{+ Bespoke}&134.21&0.241&0.206&52.51&0.243&\underline{0.212}&23.70&0.251&0.214&20.69&0.252&\underline{0.216}
> \newline
> \text{+ LD3}&\underline{55.31}&\underline{0.244}&\underline{0.208}&36.85&0.249&\underline{0.212}&\underline{20.37}&\textbf{0.258}&\textbf{0.217}&\underline{19.76}&\underline{0.258}&\textbf{0.217}
> \newline
> \text{+ BézierFlow}&\textbf{54.05}&\textbf{0.245}&\textbf{0.209}&\underline{33.43}&\textbf{0.253}&\textbf{0.214}&\textbf{19.69}&0.256&\textbf{0.217}&\textbf{16.52}&\underline{0.258}&\textbf{0.217}
> \newline
> \hline
> \text{RK2}&34.95&0.244&0.208&17.89&0.255&0.214&13.33&\underline{0.259}&0.216&11.61&0.260&0.217
> \newline
> \text{+ DMN}&36.33&0.243&0.208&\underline{16.45}&\underline{0.257}&\textbf{0.216}&27.09&0.252&0.213&17.36&0.259&0.217
> \newline
> \text{+ GITS}&\textbf{31.09}&\textbf{0.251}&\textbf{0.211}&21.21&0.255&0.214&15.58&0.257&0.216&14.65&0.258&0.216
> \newline
> \text{+ Bespoke}&45.23&0.244&0.208&40.87&0.225&0.200&20.18&0.253&0.215&13.26&0.257&0.217
> \newline
> \text{+ LD3}&39.03&0.241&0.208&18.04&0.255&\underline{0.215}&\underline{12.30}&\textbf{0.260}&\textbf{0.218}&\underline{11.54}&\underline{0.261}&\underline{0.218}
> \newline
> \text{+ BézierFlow}&\underline{33.94}&\underline{0.248}&\underline{0.210}&\textbf{16.41}&\textbf{0.258}&\underline{0.215}&\textbf{12.20}&\textbf{0.260}&\underline{0.217}&\textbf{11.02}&\textbf{0.263}&\textbf{0.219}
> \newline
> \hline
> \end{array}
>
> ---
> ### W2. Stability of LPIPS-Based Training
>
> Although we did not observe any training instability or mode collapse induced by LPIPS in our experiments, where BézierFlow outperforms the baselines, we additionally conduct an ablation study on the choice of loss to address this concern. Specifically, we compare our original LPIPS-only objective with a combined LPIPS+MSE loss while keeping all other settings fixed. As shown in the table below, across both diffusion and flow models, LPIPS alone either outperforms or matches the performance of the LPIPS+MSE combination. The robustness of BézierFlow to the choice of loss is further supported by our point cloud and layout generation experiments in App. G of the revised manuscript, where the models are trained with MSE and negative mIoU, respectively, instead of LPIPS.
>
> \begin{array}{lccccc}
> \hline
> \text{}&\text{NFE=4}&\text{NFE=6}&\text{NFE=8}&\text{NFE=10}&
> \newline
> \text{Method}&\text{FID $\downarrow$}&\text{FID $\downarrow$}&\text{FID $\downarrow$}&\text{FID $\downarrow$}
> \newline
> \hline
> \text{CIFAR-10 $32 \times 32$ with EDM}&&&&&
> \newline
> \hline
> \text{iPNDM + BézierFlow}
> \newline
> \text{+ LPIPS}&\textbf{6.93}&\textbf{3.35}&\textbf{2.81}&\textbf{2.43}
> \newline
> \text{+ LPIPS + MSE}&8.29&5.77&3.35&3.08
> \newline
> \hline
> \text{CIFAR-10 $32 \times 32$ with ReFlow}&&&&&
> \newline
> \hline
> \text{RK2 + BézierFlow}
> \newline
> \text{+ LPIPS}&\textbf{13.18}&6.00&\textbf{4.31}&3.74
> \newline
> \text{+ LPIPS + MSE}&13.21&\textbf{5.96}&4.81&\textbf{3.54}
> \newline
> \hline
> \end{array}

---

> ### Author Response · Authors · 2025-11-25
> **Response to Reviewer TYFB (2/2)**
>
> ### W3. Detailed Analysis of the Differences between the Bézier SI Scheduler and Other Schedulers
>
> We appreciate the reviewer’s request for a more principled analysis of how BézierFlow differs from LD3 and Bespoke Solver. Below, we clarify these differences in more detail.
>
>
> **(a) Clarification of the Connection Between BézierFlow and Bespoke Solver**
>
> As discussed in Sec. 4.4 of the main paper, we reiterate that Bespoke Solver **independently optimizes the per-step variables $(t_s, c_s)$ and their time derivatives $(\dot{t}_s, \dot{c}_s)$**, thus breaking the intrinsic coupling between values and derivatives. This can induce a mismatch between the numerically integrated next-step prediction and the actual learned value, which in turn can cause training instabilities. On the other hand, BézierFlow models continuous functions $(\bar\alpha_s, \bar\sigma_s)$ with Bézier curves that are **$C^2$-smooth and, by construction, satisfy the key requirements of an SI scheduler: boundary conditions, monotonicity, and differentiability**. This advantage of the BézierFlow parameterization is reflected in the quantitative results summarized in Tab. 2 of the main paper.
>
> We would be glad to further clarify any specific aspects of this section that the reviewer finds unclear.
>
> **(b) Additional Theoretical Analysis of the Connection Between BézierFlow and LD3**
>
> In App. B of the revised manuscript, we formally show that the space of sampling trajectories realizable by LD3 is a subset of that of BézierFlow, thus providing a broader search space during optimization. As a result, for any objective function over sampling trajectories (e.g., our teacher-forcing objective), the optimum of BézierFlow is guaranteed to be lower than that of LD3.
>
>
>
> ---
> ### W4. Computational Costs of BézierFlow
>
> Beyond the CIFAR-10 training-time analysis in the main paper, we additionally report runtime and memory usage for all datasets and model classes. As shown in the table below, BézierFlow trains in at most 1 hour even for a 2.5B large-scale pretrained model (Stable Diffusion v3.5) at $512\times 512$ resolution, while requiring only 22GB of GPU memory. This makes the method **practical even on a single commodity GPU commonly available in research labs**. Despite this low training and memory cost, BézierFlow improves FID by large margins over the base model, e.g., **from 50.30 to 9.55 ($\approx$81% relative improvement)** at NFE=4 on CIFAR-10 and **from 8.93 to 3.13 ($\approx$64% relative improvement)** at NFE=10 on FFHQ, as shown in Tab. 1 of the main paper. We include this computation cost analysis in App. J of the revised manuscript.
>
> \begin{array}{lcccc}
> \hline
> \text{Model / Dataset} & \text{NFE=4} & & \text{NFE=10} &
> \newline
> \text{} & \text{Training Time} & \text{GPU} & \text{Training Time} & \text{GPU} \newline
> \hline
> \text{\emph{Diffusion}} & & & & \newline
> \text{CIFAR-10 32$\times$32} & 8\text{m} & 4\text{ GB} & 15\text{m} & 8\text{ GB} \newline
> \text{FFHQ 64$\times$64} & 11\text{m} & 3\text{ GB} & 18\text{m} & 7\text{ GB} \newline
> \text{AFHQv2 64$\times$64} & 11\text{m} & 3\text{ GB} & 18\text{m} & 7\text{ GB} \newline
> \hline
> \text{\emph{Flow}} & & & & \newline
> \text{CIFAR-10 32$\times$32} & 8\text{m} & 3\text{ GB} & 15\text{m} & 7\text{ GB} \newline
> \text{ImageNet 256$\times$256} & 25\text{m} & 5\text{ GB} & 45\text{m} & 8\text{ GB} \newline
> \text{MS-COCO 512$\times$512 (SD3.5)} & 1\text{h} & 21\text{ GB} & 100\text{m} & 22\text{ GB} \newline
> \hline
> \end{array}
>
> ---
> **References**
>
> **[A]** Learning Transferable Visual Models From Natural Language Supervision, Radford *et al*., ICML 2021
>
> **[B]** Pick-a-Pic: An Open Dataset of User Preferences for Text-to-Image Generation, Kirstain *et al*., NeurIPS 2023

---

### Official Review · Reviewer_wWWP · 2025-11-01

**Soundness:** 3
**Presentation:** 3
**Contribution:** 3
**Rating:** 8
**Confidence:** 3

**Summary:**

The authors propose a new method for efficient sampling with generative models in the stochastic interpolants framework [Albergo et al. 2023], such as diffusion, score-based, and flow-based models. They optimize stochastic trajectories (i.e., the path from the latent space to the data distribution), parameterized by continuous functions $\alpha(s)$ and $\sigma(s)$, so that a scheduler requiring a small number of NFEs can be distilled from a more complex one in a teacher–student manner. The student scheduler parameters $\bar{\alpha}(s)$ and $\bar{\sigma}(s)$ are modeled using degree-$n$ 1-D Bézier curves with learnable control points. Bézier polynomials are chosen because they satisfy several constraints required by scheduler functions. The authors demonstrate effectiveness on multiple diffusion and flow-based models and datasets, achieving results that are better or comparable to the state of the art for low-NFE samplers.

**Strengths:**

* The paper is well-written and easy to follow.
* The use of Bézier polynomials to optimize schedulers for low-NFE samplers is interesting and well-motivated, and the authors conduct extensive experiments on multiple models within the stochastic interpolant framework.
* In low-NFE regimes, the method achieves quality that is often superior to existing approaches while requiring minimal training cost.

**Weaknesses:**

* In the experiments section, both the “Generalizability to Unseen NFEs” and “Training Efficiency” subsections would benefit from more detailed explanations. In particular, I found the first one somewhat unclear, while the comparison in the latter subsection is a bit confusing. I suggest providing additional training-time comparisons at equal NFE settings, including against non-distillation-based methods.
* All experiments are done on image-based models, so the choice of the LPIPS loss is justified, but it limits the scope of the evaluation.
* Minor: typo at line 299.

**Questions:**

Beyond the additional details requested above, could the authors comment on potential alternatives to LPIPS when moving beyond vision tasks? Would their method retain the same advantages under different distance metrics?

---

> ### Author Response · Authors · 2025-11-25
> **Response to Reviewer wWWP (1/2)**
>
> Dear **reviewer wWWP**,
>
> We thank the reviewer for the positive assessment and constructive suggestions based on their careful reading. Specifically, we are glad that you found the **paper well written, the Bézier parameterization well motivated, and the empirical results strong in the low-NFE regime.** Below, we address each of your concerns and questions in detail.
>
> ---
> ### W1. Clarification on “Generalizability to Unseen NFEs”
>
> The goal of this experiment is to show that once BézierFlow is trained with a single NFE, it can be instantiated for arbitrary unseen NFEs at inference time by learning a _continuous_ SI scheduler $(\bar\alpha_\theta(s), \bar\sigma_\theta(s))$. For instance, Tab. 3 of the main paper shows that BézierFlow trained with $\text{NFE}=10$ generalizes to $\text{NFE}=6$ and $8$ without retraining. In contrast, prior works optimizing _discrete per-step variables_, such as LD3 and Bespoke Solver, must be trained separately for each target NFE. We have revised the manuscript to more clearly highlight the purpose of this experiment and its key distinction from prior works. We would be glad to clarify any remaining questions the reviewer may have regarding this section.
>
>
> ---
> ### W1. Extended Comparison of “Training Efficiency”
>
> Following the reviewer’s suggestion, for a more comprehensive and fair comparison of training efficiency, we report additional results at matched NFEs with varying training budgets in the table below. As shown, under the same NFEs, distillation-based approaches (CD [A] and 2-RF [B]) yield notably worse FID with the same lightweight training budget (15 minutes) and require **substantially longer** training time (2-6 days) to achieve FID comparable to BézierFlow, corresponding to roughly **200–600× more training time**. These results underscore BézierFlow’s highly training-efficient acceleration, achieving in just a few minutes the performance that prior distillation-based approaches require several days of training to reach. Note that the 15-minute performance of 2-RF is identical to that of the base pretrained model as this budget is fully spent on the data creation stage for ReFlow.
>
> We also include training time comparisons against non-distillation baselines that accelerate generation with lightweight training, including DMN, GITS, Bespoke Solver and LD3. Among these lightweight acceleration methods, BézierFlow achieves the best FID, even outperforming LD3 under the same training budget. This demonstrates that BézierFlow offers a more favorable trade-off between training efficiency and sample quality.
>
>
> \begin{array}{lccccc}
> \hline
> \text{CIFAR-10 $32 \times 32$ with EDM}&&&&&
> \newline
> \hline
> \text{}&\text{NFE=6}&&\text{NFE=8}&&
> \newline
> \text{Method}&\text{FID $\downarrow$}&\text{Training Time $\downarrow$}&\text{FID $\downarrow$}&\text{Training Time $\downarrow$}
> \newline
> \hline
> \text{CD}&359.59&15\text{m}&343.59&15\text{m}
> \newline
> \text{CD}&4.24&6\text{ days}&3.95&6\text{ days}
> \newline
> \text{CD}&2.82&8\text{ days (A100)}&2.79&8\text{ days (A100)}
> \newline
> \hline
> \text{iPNDM}&9.84&-&5.30&-
> \newline
> \text{+ DMN}&9.33&5\text{s}&4.82&5\text{s}
> \newline
> \text{+ GITS}&6.80&30\text{s}&4.07&30\text{s}
> \newline
> \text{+ LD3}&4.42&10\text{m}&2.93&13\text{m}
> \newline
> \text{+ BézierFlow}&3.35&10\text{m}&2.81&13\text{m}
> \newline
> \hline
> \end{array}
>
> \begin{array}{lccccc}
> \hline
> \text{CIFAR-10 $32 \times 32$ with ReFlow}&&&&&
> \newline
> \hline
> \text{}&\text{NFE=6}&&\text{NFE=8}&&
> \newline
> \text{Method}&\text{FID $\downarrow$}&\text{Training Time $\downarrow$}&\text{FID $\downarrow$}&\text{Training Time $\downarrow$}
> \newline
> \hline
> \text{RK2}&12.12&-&9.17&-
> \newline
> \text{+ 2-RF}&12.12&15\text{m}&9.17&15\text{m}
> \newline
> \text{+ 2-RF}&5.69&2\text{ days}&5.45&2\text{ days}
> \newline
> \text{+ 2-RF}&3.74&8\text{ days (A100)}&3.68&8\text{ days (A100)}
> \newline
> \hline
> \text{+ DMN}&51.99&5\text{s}&21.43&5\text{s}
> \newline
> \text{+ GITS}&11.84&30\text{s}&8.77&30\text{s}
> \newline
> \text{ + Bespoke}&64.87&30\text{m}&16.67&30\text{m}
> \newline
> \text{+ LD3}&13.82&10\text{m}&6.26&13\text{m}
> \newline
> \text{+ BézierFlow}&6.00&10\text{m}&4.31&13\text{m}
> \newline
> \hline
> \end{array}

---

> > ### Author Response · Authors · 2025-11-25
> > **Response to Reviewer wWWP (2/2)**
> >
> > ### W2,Q1. Generalizability of BézierFlow across Domains and Distance Metrics.
> >
> > To demonstrate the generality of BézierFlow and provide a more comprehensive evaluation, we include experiment results beyond image generation, using alternative distance metrics. Specifically, we conduct two additional experiments with task-appropriate losses as the distance metric in Eq. 5 of the main paper: 3D point cloud generation with MSE loss and layout generation with negative mIoU. As shown, BézierFlow offers substantial improvement over the base solvers in low-NFE regime, indicating that BézierFlow is a general acceleration approach, robust across different domains and distance metrics.
> >
> > More details on the experiment setup, along with additional quantitative and qualitative results, are provided in App. G of the revised manuscript.
> >
> >
> > **(a) Unconditional 3D Point Cloud Generation**
> >
> > \begin{array}{lccccccccccccc}
> > \hline
> > \text{Method}
> > &&\text{NFE=4}&&&\text{NFE=6}&&&\text{NFE=8}&&&\text{NFE=10}&&
> > \newline
> > \text{}
> > &\text{CD-MMD}\downarrow&\text{CD-COV}\uparrow&\text{JSD}\downarrow
> > &\text{CD-MMD}\downarrow&\text{CD-COV}\uparrow&\text{JSD}\downarrow
> > &\text{CD-MMD}\downarrow&\text{CD-COV}\uparrow&\text{JSD}\downarrow
> > &\text{CD-MMD}\downarrow&\text{CD-COV}\uparrow&\text{JSD}\downarrow
> > \newline
> > \hline
> > \text{UniPC}&2.50&3.21&0.46&1.25&8.89&0.30&0.95&17.03&0.25&0.79&20.25&\underline{0.22}
> > \newline
> > \text{+ DMN}&\underline{1.10}&16.79&\underline{0.27}&\underline{0.68}&\textbf{26.42}&\textbf{0.23}&1.50&15.06&0.32&\underline{0.67}&19.51&0.27
> > \newline
> > \text{+ GITS}&5.32&9.52&0.56&9.14&0.74&0.63&1.20&20.25&0.31&0.90&16.30&0.23
> > \newline
> > \text{+ LD3}&1.20&\textbf{21.23}&\textbf{0.24}&1.16&20.74&\underline{0.24}&\underline{0.80}&\underline{21.48}&\underline{0.25}&0.91&\underline{21.23}&0.23
> > \newline
> > \textbf{+ BézierFlow (Ours)}&\textbf{0.88}&\underline{18.77}&0.29&\textbf{0.59}&\underline{22.72}&\textbf{0.23}&\textbf{0.58}&\textbf{23.45}&\textbf{0.24}&\textbf{0.53}&\textbf{23.70}&\textbf{0.21}
> > \newline
> > \hline
> > \text{iPNDM}&\underline{1.17}&14.81&\textbf{0.27}&0.91&16.54&\underline{0.23}&0.78&\underline{23.46}&\textbf{0.21}&0.67&\textbf{26.67}&\textbf{0.20}
> > \newline
> > \text{+ DMN}&1.18&\textbf{19.26}&\underline{0.29}&\underline{0.63}&\textbf{24.69}&\textbf{0.22}&1.74&6.17&0.35&\underline{0.65}&20.49&0.22
> > \newline
> > \text{+ GITS}&3.22&7.65&0.44&3.59&3.21&0.48&3.99&1.73&0.49&2.73&3.95&0.41
> > \newline
> > \text{+ LD3}&2.40&13.33&0.34&0.89&18.52&0.25&\underline{0.77}&19.01&0.25&0.70&22.72&0.22
> > \newline
> > \textbf{+ BézierFlow (Ours)}&\textbf{0.85}&\underline{18.52}&\underline{0.29}&\textbf{0.58}&\underline{22.73}&\underline{0.23}&\textbf{0.57}&\textbf{24.44}&\underline{0.23}&\textbf{0.56}&\underline{24.52}&\underline{0.21}
> > \newline
> > \hline
> > \end{array}
> >
> > **(b) Unconditional Layout Generation**
> >
> > \begin{array}{lccccccccccccc}
> > \hline
> > \text{Method}
> > &&\text{NFE=4}&&&\text{NFE=6}&&&\text{NFE=8}&&&\text{NFE=10}&&
> > \newline
> > \text{}
> > &\text{FID}\downarrow&\text{Alignment}\downarrow&\text{Overlap}\downarrow
> > &\text{FID}\downarrow&\text{Alignment}\downarrow&\text{Overlap}\downarrow
> > &\text{FID}\downarrow&\text{Alignment}\downarrow&\text{Overlap}\downarrow
> > &\text{FID}\downarrow&\text{Alignment}\downarrow&\text{Overlap}\downarrow
> > \newline
> > \hline
> > \text{RK1}&55.88&0.40&0.60&22.75&0.35&0.56&11.66&0.30&0.54&7.93&0.27&0.52
> > \newline
> > \text{+ DMN}&178.35&0.55&1.08&88.40&0.69&0.70&26.27&0.37&\underline{0.46}&10.96&0.33&\textbf{0.46}
> > \newline
> > \text{+ GITS}&41.08&0.37&0.57&12.84&0.35&\textbf{0.47}&7.32&0.29&\textbf{0.45}&5.90&0.27&\textbf{0.46}
> > \newline
> > \text{+ Bespoke}&213.61&0.92&1.01&201.20&0.88&0.67&168.49&0.63&0.59&171.11&0.63&0.56
> > \newline
> > \text{+ LD3}&\textbf{19.51}&\textbf{0.32}&\underline{0.54}&\underline{8.36}&\underline{0.28}&0.51&\underline{5.03}&\textbf{0.23}&0.48&\underline{3.70}&\underline{0.23}&\underline{0.47}
> > \newline
> > \text{+ BézierFlow}&\underline{32.78}&\underline{0.35}&\textbf{0.53}&\textbf{7.10}&\textbf{0.26}&\underline{0.47}&\textbf{3.86}&\underline{0.25}&0.48&\textbf{2.96}&\textbf{0.21}&0.49
> > \newline
> > \hline
> > \text{RK2}&143.90&0.67&0.65&73.91&0.47&0.49&35.84&0.38&0.51&20.80&0.34&0.51
> > \newline
> > \text{+ DMN}&142.40&0.66&\textbf{0.35}&88.15&0.49&\underline{0.46}&63.57&0.37&\textbf{0.42}&56.23&0.35&\textbf{0.43}
> > \newline
> > \text{+ GITS}&\textbf{102.11}&\textbf{0.49}&\underline{0.42}&51.62&\underline{0.37}&0.48&27.84&\underline{0.32}&0.50&\underline{8.25}&\textbf{0.22}&\underline{0.47}
> > \newline
> > \text{+ Bespoke}&\underline{126.80}&\underline{0.61}&0.48&187.62&0.86&\textbf{0.37}&32.99&0.38&0.54&21.54&0.36&0.50
> > \newline
> > \text{+ LD3}&162.98&0.62&0.47&\underline{42.82}&\underline{0.37}&0.48&\textbf{12.57}&\textbf{0.26}&0.48&8.39&0.27&\underline{0.48}
> > \newline
> > \text{+ BézierFlow}&142.34&0.63&0.57&\textbf{39.17}&\textbf{0.35}&0.52&\underline{25.51}&0.37&0.49&\textbf{7.18}&\underline{0.25}&0.49
> > \newline
> > \hline
> > \end{array}
> >
> > ---
> > **References**
> >
> > [A] Consistency Models, Song *et al*., ICML 2023
> >
> > [B] Learning to Generate and Transfer Data with Rectified Flow, Liu *et al*., ICLR 2023

---

> > > ### Comment · Reviewer_wWWP · 2025-11-27
> > >
> > > I thank the authors for their thorough responses and for providing additional experiments. The generalisation ability of BeziérFlows to unseen NFEs and its training properties are now clearer. For these reasons, I confirm my previous positive score.

---

> ### Author Response · Authors · 2025-11-27
> **Response to Reviewer wWWP**
>
> Dear Reviewer wWWP,
>
> We are glad to hear that our rebuttal addressed your remaining concerns, and we sincerely appreciate your final positive assessment of our work.
>
> Best,
>
> Authors

---

### Author Response · Authors · 2025-11-25
**Common Response**

Dear reviewers and AC,

We sincerely appreciate your valuable time and effort spent reviewing our manuscript.

We have updated the manuscript in response to the reviewers’ comments. The main updates to the appendix are:

* Additional theoretical analysis of relation between LD3 and BézierFlow in Sec. B
* BézierFlow at extremely low NFEs (NFE=1, 2, and 3) in Sec. E.1
* BézierFlow on other domains, including 3D point cloud generation and layout generation, in Sec. G
* Zero-shot cross-dataset transfer of BézierFlow in Sec. I

Please refer to the revised manuscript for details, alongside our responses.

---

### Meta-Review · Area_Chair_6h8T · 2026-01-04

**Summary:**

Three reviewers raised several concerns regarding the paper’s methodology and experiments:

1) Reviewers requested more clarification on the paper’s difference from related works, its certain technical details, and its training setups.

2) Reviewers suggested a more comprehensive analysis and evaluation of the method, including reporting more performance metrics, comparing it with more baselines, and evaluating the method on larger-scale and more challenging datasets.

3) Reviewers also questioned the method’s application in other problem domains besides vision tasks.

**Reviewer Concerns:**

1) The rebuttal has provided sufficient clarification on these questions. In particular, its clarifications on the method’s generalizability, its training efficiency, and its distinction from related work have been well received by the two positive reviewers. I think the rebuttal has properly addressed this concern.

2) The rebuttal has added additional experiments in response to these suggestions and concluded that the proposed model remains effective in various settings. After reading the reviews and the rebuttal, I think these new experiments are helpful for readers to better understand the efficacy of the proposed method, and I consider this concern properly addressed.

3) The rebuttal responded with additional results from 3D point cloud and layout generation tasks and concluded that the proposed method is competitive across these problem domains. These new experiments have resolved the reviewer’s concern.

**Reviewer Scores:**

I think the negative reviewer would probably become supportive of the work, as the rebuttal has provided additional experiments to address each weakness. The other reviewers were initially positive and have confirmed their support through the early discussion. Based on all these factors, I think the three reviewers would have reached a consensus to accept this paper had they been able to participate in a full discussion.

---

### Decision · Program_Chairs · 2026-01-26

Accept (Poster)